# Large Granular Lymphocytic Leukemia: Clinical Features, Molecular Pathogenesis, Diagnosis and Treatment

**DOI:** 10.3390/cancers16071307

**Published:** 2024-03-27

**Authors:** Fauzia Ullah, Mariam Markouli, Mark Orland, Olisaemeka Ogbue, Danai Dima, Najiullah Omar, Moaath K. Mustafa Ali

**Affiliations:** 1Department of Translational Hematology and Oncology Research, Lerner Research Institute, Cleveland Clinic, Cleveland, OH 44915, USA; orlandm5@ccf.org (M.O.); ogbueo@ccf.org (O.O.); dimad@ccf.org (D.D.); najiullahomar@gmail.com (N.O.); mustafm8@ccf.org (M.K.M.A.); 2Department of Internal Medicine, Boston Medical Center, Boston University School of Medicine, Boston, MA 02118, USA; 3Department of Hematology and Medical Oncology, Taussig Cancer Institute, Cleveland Clinic Foundation, Cleveland, OH 44915, USA

**Keywords:** large granular lymphocytic leukemia, molecular pathogenesis, natural killer LGL leukemia, STAT3, targeted therapies

## Abstract

**Simple Summary:**

The insights gleaned from available evidence contribute to our understanding that clonal LGL arises from chronic antigenic stimulation by a virus. The findings discussed here provide insights into the molecular pathogenesis of the disease. The current treatment is immunosuppressive monotherapy, and there is no standard first-line regimen for the disease. Unfortunately, these therapies have limited efficacy in eradicating the LGL clone and sustain satisfactory long-term remission. The discovery of novel therapeutic targets and large clinical trials could potentially lead to improved response.

**Abstract:**

Large granular lymphocytic (LGL) leukemia is a lymphoproliferative disorder characterized by persistent clonal expansion of mature T- or natural killer cells in the blood via chronic antigenic stimulation. LGL leukemia is associated with specific immunophenotypic and molecular features, particularly *STAT3* and *STAT5* mutations and activation of the *JAK-STAT3*, Fas/Fas-L and *NF-κB* signaling pathways. Disease-related deaths are mainly due to recurrent infections linked to severe neutropenia. The current treatment is based on immunosuppressive therapies, which frequently produce unsatisfactory long-term responses, and for this reason, personalized approaches and targeted therapies are needed. Here, we discuss molecular pathogenesis, clinical presentation, associated autoimmune disorders, and the available treatment options, including emerging therapies.

## 1. Introduction

Large granular lymphocyte (LGL) leukemia is a rare clonal disease characterized by the proliferation of large granular lymphocytes, leading to peripheral and marrow lymphocytic infiltration and cytopenia, most notably neutropenia. LGLs are a distinct lymphoid subtype that are larger than typical circulating lymphocytes (15–18 µm) and have characteristic azurophilic granules containing acid hydrolases. LGLs constitute 10–15% of normal peripheral mononuclear cells and arise from two major lineages. Approximately 85% are CD3 and CD57 positive and CD56 negative cytotoxic T cell, while the remaining 15% are CD3 negative and CD56 positive natural killer (NK) cell [1,2].

The classification landscape for neoplastic disorders was updated by two prominent working groups in 2022, namely the International Consensus Classification (ICC) and the World Health Organization (WHO)-HAEM5, the fifth edition of the WHO classification, which now stand as the two currently available options for both lymphoid and myeloid neoplasms. The overarching categories of lymphoid malignancies in both the ICC and WHO-HAEM5 have not undergone any substantial modifications in comparison to the prior WHO-HAEM4R. However, both schemes incorporated changes to the nomenclature and categorization of some lymphoid neoplasms. In general, the WHO-HAEM5 classifies mature T-cell malignancies and NK-cell neoplasms into nine distinct categories. One of the nine categories is mature T-cell and NK-cell leukemias, which includes six entities representative of T- and NK-cell proliferation that primarily present as leukemic disease and include the following: T-cell large granular lymphocyte (T-LGL) leukemia, NK-cell large granular lymphocyte (NK-LGL) leukemia, and aggressive NK-cell leukemia (ANKL). In contrast, the ICC considered all entities as a single group of mature T-cell and NK-cell neoplasms [3,4,5,6]. While the term chronic lymphoproliferative disorder of NK cells is kept in the ICC criteria, it was replaced with NK-LGL leukemia in WHO-HAEM5. 

Emerging evidence underscores the clinical and phenotypic importance of specific mutations in T-LGL leukemia, particularly activator of transcription 3 (*STAT3*), which is prevalent in CD8+ T-LGL and gamma/delta T-LGL leukemias and has been shown to correlate with neutropenia and poor overall survival. Conversely, the *STAT5b* mutation found in approximately 30% of CD4+ T-LGL cases lacks prognostic impact, while rare CD8+ T-LGLs have a poor prognosis. The re-classification of chronic lymphoproliferative disorders of NK cells as NK-LGL leukemia reflects recent evidence demonstrating monoclonal or oligoclonal expansion of NK cells with similarities to T-LGL leukemia. These insights provide a foundation for a deeper understanding of the genetic landscape and potential therapeutic targets in T-LGL and NK-cell leukemia [7,8,9,10,11].

Most patients with LGL are clinically asymptomatic and tend to have a chronic clinical course. While the majority of patients eventually need treatment due to severe or symptomatic neutropenia, anemia and/or lymphopenia, there is no standard established therapy currently, and treatment is based on immunosuppressive therapies that produce unsatisfactory long-term response [12,13,14]. Active research is underway to better understand the molecular pathogenesis of LGL leukemia and to develop effective therapeutic targeted regimens. This review will discuss molecular pathogenesis, clinical manifestations, associated autoimmune disorders, and management, including emerging therapies.

## 2. Clinical Manifestations of LGL Leukemia

LGL leukemia comprises approximately 2–5% of all chronic lymphoproliferative diseases diagnosed in the US and Europe, with a slightly high prevalence of 5–6% observed in the Asian population [15]. The average incidence of LGL leukemia in Europe and North America is 0.2–0.72 million persons per year [16,17]. The disease is most commonly diagnosed in elderly patients, with a median age at presentation of 55–60 years [16,18]. It equally affects men and women, but the diagnosis in women is often made at a younger age [16]. The median overall survival is 9 to 10 years. Notably, only 10% of patients are younger than 40 years, and pediatric cases are sporadic [2]. The presence of significant comorbidities and age >60 years at the time of diagnosis have been shown as independent predictors of poor survival [16]. The two subtypes of chronic LGL proliferation are described as follows: T-LGL and NK-LGL.

### 2.1. T-LGL Leukemia

T-LGL is associated with various types of cytopenias. Even though most T-LGL leukemia cases follow an indolent course, up to two-thirds of patients will become symptomatic, and approximately 80% and 45% of symptomatic patients will develop neutropenia and severe neutropenia (absolute neutrophil count less than 0.5 × 10^9^/L), respectively [1]. Neutropenia is the most frequent cytopenia seen in LGL leukemia, increasing the risk of bacterial infections. Severe neutropenia complicated by infection is commonly observed in Western countries, whereas severe anemia is more frequently seen in Japan and China [1,19,20,21,22,23,24,25,26,27,28,29]. The mechanism of neutropenia is multifactorial but most likely involves the secretion of pro-inflammatory cytokines, possibly through a Fas/Fas-ligand (Fas-L) dependent mechanism [2,30,31]. T-LGLs express Fas-L, while mature neutrophils express Fas, allowing for Fas/Fas-L-mediated neutrophilic apoptosis [32]. Several studies had previously reported the presence of serum anti-neutrophil autoantibodies in some patients, but most of them were anti-HLA antibodies instead of true anti-neutrophil autoantibodies, and are thus less likely to be the actual cause of neutropenia [32].

Anemia is thought to be due to a T-LGL-mediated suppression of erythroid progenitors [33,34], although the exact mechanism is unclear. It has been suggested that T-LGLs either kill the erythroid progenitor cells directly or suppress them indirectly through humoral mediator release [35]. Immune thrombocytopenia and myelodysplastic syndrome (MDS) have been documented as well [36]. It needs to be noted that the frequent coexistence of T-LGLs with bone marrow (BM) failure syndromes, such as aplastic anemia (AA), paroxysmal nocturnal hemoglobinuria (PNH), and MDS [36,37,38], suggest a potential etiological relationship between these entities. A study of 367 patients with MDS and 140 patients with AA were screened for the presence of concomitant *STAT3* mutations and LGL [39]. The result demonstrated that *STAT3* clones can be found not only in AA and MDS patients with concomitant LGL but also in unexpected cases (7% AA, 2.5% MDS). Remarkably, AA patients with concomitant *STAT3* mutations were characterized by a better response to immunosuppression, particularly in those with the presence of HLA-DR15. On the other hand, patients with MDS and *STAT3* mutated clones demonstrated a lower degree of BM cellularity but also increased chances of harboring chromosome 7 abnormalities. The authors concluded that LGL clones with *STAT3* mutation can be found in a small proportion of acquired BM failure syndromes, such as AA and MDS, and this mechanism might also be involved in associated autoimmune diseases, including rheumatoid arthritis (RA). In addition, previous studies have indicated that 30–46% of BM failure and LGL leukemia patients have increased levels of antibodies to *BA21* protein, which is found in the transmembrane region of the human T cell lymphotropic virus (HTLV-1) envelope [40]. This suggests that, in some cases, LGL leukemia, AA, MDS, and PNH might share a common pathogenesis [8]. 

Initial manifestations of the disease are mainly related to neutropenia and include fever with recurrent bacterial infections involving the skin, oropharynx, and perirectal areas. Severe sepsis or pneumonia may also occur, but opportunistic infections are uncommon. The reported frequency of recurrent infections varies in the literature, ranging from 15% to 39%. On the other hand, profound and persistent neutropenia without infection may also be observed over a very long period in some patients. Thrombocytopenia is observed in around 20% of patients with T-LGL; however, severe thrombocytopenia is rare, occurring in approximately 1%, and is typically linked to the suppression of megakaryopoiesis by clonal LGL cells or related marrow diseases [41]. Fatigue and B symptoms (fever, night sweats, weight loss) are common, as seen in 20–30% of cases. The frequency of splenomegaly and hepatomegaly ranges from 20% to 50% and 10% to 20%, respectively, whereas lymphadenopathy is rare [15]. A few case series have identified pulmonary hypertension as the presenting manifestation [42]. Neuropathy and hemophagocytic syndrome may very rarely occur as well [2,22]. 

### 2.2. NK-LGL Leukemia

The clinical presentation of NK-LGL is similar to that of T-LGL leukemia. It is an indolent disease subtype with a good prognosis that is usually detected in routine blood studies with persistently elevated circulating LGLs. NK-LGL patients can present with neutropenia, anemia, or thrombocytopenia and have an underlying array of various autoimmune conditions, but less frequently than T-LGL leukemia patients [43]. The differential diagnosis of NK-LGL leukemia also includes aggressive NK-LGL leukemia, a rare LGL proliferation characterized by poor prognosis and younger age of presentation, with a median age of 39 years at diagnosis, usually affecting patients of Asian descent [44]. It is associated with Epstein–Barr virus (EBV) and represents 5% of LGL proliferation. Patients commonly experience fulminant B symptoms, hepatosplenomegaly, and a wide range of cytopenias [45].

## 3. Pathogenesis

The exact cause of the clonal proliferation of LGL is not known; however, the initial activating step is suggested to be antigen-driven, resulting in oligoclonal LGL expansion [11,12]. It is well known that chronic inflammation from deregulated cytokine production can increase the risk of malignant transformation of normal host cells [46]. Patients with LGL leukemia have increased serum levels of soluble interleukin-15 (IL-15) Rα, IL-2/15Rβγ and the membrane-bound form of IL-15 in leukemic blasts [47]. IL-15 is a pro-inflammatory cytokine, playing a crucial role in the development, survival, proliferation and activation of several lymphocyte lineages through diverse signaling pathways. Chromosomal instability, microRNA alterations and DNA hypermethylation caused by chronic IL-15 exposure may lead to the generation of *STAT3/STAT5b* mutations, which then cause aberrant activation of the signaling pathways, contributing to clonal proliferation. Mishra et al. demonstrated chromosomal instability and leukemic transformation of a wild-type mouse LGL after chronic exposure to IL-15 alone [48]. The authors concluded that chronic exposure of cells to IL-15 led to overexpression of aurora kinases, *AurkA* and *AurkB*, resulting in centrosome abnormalities. Since these kinases are regulated by *Myc*, the proto-oncogene, increased *Myc* expression via the activation of *NF-κB* was noted.

LGLs are activated through antigen recognition, then undergo significant expansion and eventually death by apoptosis after antigen clearance. In LGL leukemia, however, these LGLs persist [49]. There is abnormal regulation of LGL leukemic cells through the pathways involved in activation-induced cell death (AICD), leading to expanded effector memory cytotoxic T cell population by chronic persistent antigen exposure that then leads to the *STAT3* activation mutation, which is seen in 30–40% of patients [9,10,49]. The following sections describe the molecular pathogenesis of LGL leukemia, and a summary of the pathways reported in the literature [15,50,51] is shown in Figure 1.

### 3.1. Cytokine Signaling Dysregulation 

The pathogenesis of LGL leukemia is very complex and involves the dysregulation of various cytokines and molecular pathways, with the cytokine profile between the different disease subtypes being largely similar [45,52]. IL-15 and platelet-derived growth factor (PDGF) have been identified as key survival signaling switches that influence most of the dysregulations that occur in this disease [47,48,53,54,55,56]. IL-15 utilizes three distinct receptor chains to transmit signals and induce effects [48]. IL-15, expressed by antigen-presenting cells, binds to IL-2Rβγ and IL-15Rβγ heterodimers, activating *JAK1*, which subsequently phosphorylates *STAT3* and *JAK3/STAT5* via its β and γ chains, respectively. Phosphorylated *STAT3* and *STAT5* proteins form heterodimers that translocate to the nucleus, where they facilitate the transcriptional activation of anti-apoptotic proteins *Bcl-2, Bcl-xL* and proto-oncogenes. A second pathway involves the recruitment of an adaptor protein, Shc, which binds to a phosphorylated site on the IL-2/15Rβ chain, resulting in the *PI3K* and *AKT* signaling pathways increasing cell proliferation and survival. Similarly, a third signaling pathway involves the activation of the *RAS-RAF* pathway, which, in turn, activates *MAPK* to facilitate cellular proliferation [57,58,59,60]. IL-15 is a potent chemoattractant, leading to the recruitment of T- and NK cells at the site of its production [61]. Similarly, PDGF functions as another central contributor to LGL leukemia pathogenesis, with increased circulating PDGF levels in LGL leukemia patients, as well as activated PDGF downstream targets, such as *PI3K* and *AKT/ERK* [54,62]. The pharmacological disruption of PDGF signaling leads to decreased downstream target activity and increased LGL leukemic cell apoptosis.

Additionally, IL-2 expression is increased in LGL leukemic cells, which normally initiate T cell activation and guide the cell towards apoptosis through AICD [63,64]. The dysregulation of this pathway promotes tumor cell resistance to Fas-mediated apoptosis and IL-2 signaling-mediated activation of the *JAK/STAT*, *NF-kB* and *MAPK* pathways, leading to tumor cell proliferation and survival [64,65]. LGL leukemia patients are also characterized by increased IL-6 levels, mainly caused by persistent *STAT3* stimulation, and therefore, the inhibition of IL-6 signaling leads to a decrease in phosphorylated *STAT3*, thereby reducing LGL leukemic cell survival [66,67]. However, Kim et al. demonstrated that increased cytokine levels in LGL leukemia are independent of *STAT3* mutation status [68]. IL-6 further promotes *JAK/STAT* and *RAS/MAPK* signaling. IL-12 can further act as a co-stimulatory cytokine and help increase LGL cell proliferation through *JAK/STAT* signaling together with CD3 activation [69,70]. 

Other cytokines, such as IL-8, IL-10 and TNF-α, appear to be increased in LGL leukemia cultures as well, when compared to healthy controls and seem to inhibit hematopoiesis [63]. In particular, IL-8 leads to neutrophilic extravasation, which could possibly explain the neutropenia and other concurrent autoimmune diseases that patients with LGL leukemia experience [71]. Interferon gamma-inducible protein 10 (IP-10) recruits lymphocytes and also contributes to the neutropenia that characterizes this disease. On the other hand, epidermal growth factor (EGF) is decreased in LGL leukemia [50].

### 3.2. Molecular Pathway Dysregulation

Regarding the dysregulation of molecular pathways, LGL cells have been found to display increased activated signal transducer and *STAT3* levels; therefore, treatment with in vitro *JAK*-selective tyrosine kinase inhibitor, *AG-490*, induced apoptosis in LGL leukemia and decreased *STAT3*-DNA binding activity [72]. Remarkably, recurrent gain-of-function somatic mutations have recently been discovered in the Src homology (SH) 2 domain of *STAT3* in 27–40% of patients with T-LGL leukemia and 30% of NK-LGL patients, which increase the gene’s transcriptional activity, leading to constitutive *STAT3* activation and gene dysregulation downstream of *STAT3* [9,10]. Patients with *STAT3* mutations frequently present with neutropenia, symptomatic disease, rheumatoid arthritis and often have refractory/relapsed disease to multiple different treatments compared to patients without these mutations. Moreover, activating mutations in the SH2 domain of *STAT5b* have been identified in 2% of NK-LGL leukemia patients, who are characterized by a more aggressive and fatal clinical course, unlike typical LGL leukemia [73]. Activating *STAT5b* mutations have also been found in 55% of CD4+/CD8−/TCRαβ+ T-LGL leukemia patients but are very rare among patients with CD4−/CD8+ T-LGL leukemia or NK-LGL [74]. Similarly, CD3+/CD8+/CD57+ and CD3+/CD4+/CD8^dim^/CD57+ phenotypes have been reported in T-LGL with *STAT3* mutation, while CD3+/CD4+/CD57+ and CD3+/CD56+ are found in *STAT5b* mutation [75]. One study demonstrated that the disease course in the CD4+ T-LGL leukemia cohort is indolent, and none of the patients with *STAT5b* mutations appeared to require therapy during the observation period [74]. Of note, the acquisition of *STAT3/STAT5b* mutations occurs later than the establishment of clonality in the natural history of the disease [76]. *STAT3* and *STAT5b* mutations can thereby be used as molecular markers for diagnosing the disease while also serving as potential therapeutic targets with *STAT3* and *STAT5b* inhibitors that are already in development [45]. 

The process of Fas-mediated apoptosis is also impaired, which may support LGL accumulation since leukemic LGLs have increased Fas/Fas-L levels and are resistant to Fas/Fas-L mediated apoptosis [45,77]. Fas and Fas-L signaling normally plays a fundamental role in immune system regulation by inducing apoptosis. Fas-L binding by Fas results in death-inducing signaling complex (DISC) formation, which activates caspase-dependent apoptosis. Through this mechanism, cytotoxic T cells, including LGLs, can induce cell death of foreign or infected cells. Fas/Fas-L signaling also aids in T cell homeostasis by eliminating activated CTLs as the infection subsides, which are typically apoptosis-resistant during the activation and clonal expansion process. This process, known as AICD, limits the risk of excessive immune response and T cell reaction to self-antigens. Leukemic LGLs are resistant to Fas/Fas-L mediated AICD, even though they express high levels of Fas-L and do not appear to have any mutations in the Fas receptor gene [77]. FADD and c-FLIP are overexpressed in leukemic LGLs and subsequent overactivation of the Src family kinases results in the constitutional activation of the *PI3K-AKT* pathway, which leads to the inhibition of DISC formation through *MCL-1*, an anti-apoptotic protein that prevents apoptosis [45]. In this context, *NF-kB* appears to be activated downstream of *AKT* in leukemic LGLs, promoting anti-apoptotic *Bcl-2* protein expression. Imbalances in sphingolipid levels have also been discovered in LGL leukemia, with anti-apoptotic sphingosine-1-phosphate (S1P) being elevated and pro-apoptotic ceramide being decreased. 

### 3.3. Response to Viral Antigens

HTLV-1 has been associated with certain types of T-cell malignancies, but its direct involvement in LGL leukemia, including both T-LGL and NK-LGL, is not well established. While viral infections, including EBV, have been implicated in some cases of NK-LGL leukemia [78,79], the role of HTLV-1 remains a subject of research. HTLV-1 retroviral infection in LGL leukemia pathogenesis has primarily been suggested by serologic studies, showing cross-reactivity to viral epitopes, such as the envelope protein *BA21*, in 30% to 50% of patients [12,80,81,82,83,84]. Association with HTLV-2 has also been occasionally described [85,86]. Furthermore, two T-LGL leukemia cases linked to B-indolent lymphoma were reported in Hepatitis C-infected patients and were successfully treated with antiviral therapy [87].

## 4. Associated Autoimmune and Bone Marrow Failure Disorders

The association between LGL and other autoimmune disorders is well established, with approximately one-third of those with LGL leukemia presenting with a concomitant autoimmune disease or bone marrow disorders, commonly rheumatoid arthritis and pure red cell aplasia (PRCA) [43,88]. Notably, CD3+/CD8+/CD57+ LGL leukemia is often associated with autoimmune disorders, while CD3+/CD4+/CD57+ LGL leukemia is associated with monoclonal B lymphocytosis [75].

### 4.1. LGL and Autoimmune Disorders

While multiple autoimmune disorders have been reported with LGL, including systemic lupus erythematosus and Hashimoto’s thyroiditis, the most common are rheumatoid arthritis and Sjogren syndrome.

#### 4.1.1. Rheumatoid Arthritis

Rheumatoid arthritis is a chronic inflammatory disorder that leads to joint destruction and extra-articular involvement. Serologic abnormalities (positive rheumatoid factor, anti-ccp antibodies, and polyclonal hypergammaglobulinemia) are also common. While the exact cause of RA is unknown, one hypothesis is that polyclonal CD8+ T-cell expansion targeting neutrophils leads to downstream cytotoxic granule release. This induces leukotoxic hypercitrullination, resulting in neoepitope formation secondary to cell lysis [89]. Sixty percent of those with RA have been found to have polymorphisms in HLA-DRB1 with a shared epitope in the hypervariable region of DRB1, leading to poor T cell antigen presentation, along with a smaller subset who present with epigenetic changes, such as PTPN11 hypermethylation, promoting the overexpression of fibroblast-like synoviocytes [90]. Additionally, given the increased incidence of risk factors, such as smoking, the role of exposomics is being explored [91]. 

Approximately 17–36% of patients with T-LGL also have RA, which is typically diagnosed prior to LGL and more commonly noted with T-LGL than NK-LGL [12,92]. The exact correlation between the two disorders has not been fully elucidated, but several hypotheses have been proposed.

One hypothesis is that RA could be a paraneoplastic manifestation of T-LGL. The CD8+ T-cells in T-LGL drive lytic neutrophil cell death through the leukotoxic hypercitrullination pathway, catalyzed by perforin, thus creating autoantigens that precipitate RA [93]. While the exact trigger that facilitates cell lysis is yet to be elucidated, the neutropenia of T-LGL may be the result of elevated soluble Fas-L (sFas-L) levels, a cleaved product of Fas, and thus, increased apoptosis; this could explain the overlap between Felty syndrome and LGL [94]. Another proposed mechanism is that RA is a cause of LGL, where chronic autoantigen exposure to CD8+ T-cells in RA leads to monoclonal expansion with acquired somatic gene rescue in *STAT3* and alternative proliferator genes, resulting in progression to LGL [95]. Finally, treatment with TNF inhibitors in RA is significantly associated with the clonal T-LGL expansion of cells expressing CD3, CD56 and γδ TCRs and has been thought to precipitate T-LGL in some cases [96]. With the varying hypotheses on why LGL and RA often co-present, further research is still needed to clarify the link between the two diseases.

#### 4.1.2. Sjogren’s Syndrome

Sjogren’s syndrome is an autoimmune disorder characterized by the infiltration of autoreactive T- and B cells into the salivary and lacrimal glands, resulting in inflammation and tissue damage. Few cases of Sjogren’s syndrome occurring concurrently with T-LGL leukemia have been reported in the literature, and the causal relationship between Sjogren’s syndrome and T-LGL leukemia is unclear [97]. One proposed mechanism involves the activation of IL-15. This cytokine is overexpressed in the salivary glands of patients with Sjogren’s and may contribute to the activation and expansion of LGLs [98,99]. As discussed previously, IL-15 promotes the expansion of LGLs through *JAK-STAT* and *PI3K-AKT* signaling pathways and the *Myc* pathway [100,101]. 

### 4.2. LGL and Bone Marrow Failure Disorders 

Bone marrow failure disorders in LGL include neutropenia, PRCA, autoimmune hemolytic anemia (AIHA), AA, PNH, MDS and other lymphoproliferative disorders. Only neutropenia, PRCA and AIHA will be discussed here.

#### 4.2.1. Neutropenia

In approximately 10–20% of patients with LGL leukemia, neutropenia is a common clinical manifestation [12]. One potential mechanism already discussed is elevated Fas-L levels on LGLs that facilitate neutrophil apoptosis, thus leading to this presentation. Another potential mechanism involves the overproduction of TNF-α and INF-γ by LGLs, which causes bone marrow suppression and decreases the production of neutrophils [102]. In LGLs, the overproduction of TNF-α and IFN-γ can lead to the activation of the Fas-L pathway and stimulate the production of reactive oxygen species (ROS), which can cause direct damage to the BM microenvironment, worsening neutropenia [103]. Importantly, PRCA often accompanies T-LGL leukemia in Asians, while in Western countries, neutropenia is more common [25,28].

#### 4.2.2. Pure Red Cell Aplasia

PRCA is a rare BM failure syndrome characterized by a severe decrease or absence of erythrocyte precursors in the BM. The association between LGL leukemia and PRCA has been reported in several studies, including the largest study from the Mayo Clinic [28,104,105,106]. PRCA is seen in 8–19% of T-LGL leukemia patients [22,28].

Most patients with LGL leukemia and PRCA do not have lymphocytosis in their blood or bone marrow, thus making it difficult to diagnose LGL leukemia. Therefore, an evaluation with flow cytometry and T-cell gene rearrangement studies should be part of the routine work-up. Multiple immunological mechanisms have been proposed to elucidate the destruction of red blood cell precursors in RPCA, and one such mechanism similar to LGL neutropenia is the overproduction of TNF- α and IFN-γ, which can lead to the apoptosis of erythroid precursors in the BM, resulting in PRCA. LGLs express killer-cell inhibitory receptors, which inhibit their cytotoxic activity against cells expressing specific MHC class 1 antigens. Given that erythroblasts exhibit low levels of MHC class 1 molecules, they become more susceptible to LGL-mediated lysis [107]. Additionally, the production of ROS in the BM microenvironment can further exacerbate PRCA [103]. 

#### 4.2.3. Autoimmune Hemolytic Anemia

Autoimmune hemolytic anemia (AIHA) is a condition characterized by the destruction of red blood cells by autoantibodies. AIHA is a common clinical manifestation in approximately 10–15% of patients with T-LGL leukemia patients. The proposed mechanism for the development of AIHA in LGL leukemia involves the overproduction of multiple cytokines by LGLs, including IL-1, TNF-α, IFN-γ, and IL-10, which leads to the production of autoantibodies against erythrocytes [12]. Furthermore, IL-1, IFN-γ and TNF-α are known to promote hemolysis by inducing ROS production and by stimulating the complement cascade [103]. In addition, LGL cells have also been shown to express Fas-L, which can induce the apoptosis of erythrocytes via interaction with the Fas receptor [64]. 

## 5. Diagnostic Criteria for Treatment of LGL Leukemia

Clinical suspicion of LGL leukemia is warranted in scenarios involving unexplained neutropenia, recurrent infections, anemia, lymphocytosis, or autoimmune conditions. The diagnosis of LGL leukemia relies on key criteria, including clinical presentation, cellular morphology, immunophenotype, and molecular analysis. Normal LGLs exist at 0.25 × 10^9^ LGLs/L in the peripheral blood [19]. Examination of the peripheral blood morphology and evaluation of lymphocyte cell counts and immunophenotyping via flow cytometry utilizing markers such as CD3, CD4, CD5, CD8, CD16, CD27, CD28, CD45, CD56, CD57, CD62, and CD94 should be conducted [45,108,109]. Additionally, evaluation of TCR beta chain constant region (TRBC1) or TCRVβ and killer cell immunoglobulin-like receptor (KIR; CD158) expression can be assessed via flow cytometry [45,108,109]. In the case of suspected T-LGL, testing for TCR rearrangement is performed. Molecular genetic testing, including evaluation for *STAT3/STAT5* alterations, should be included in the diagnostic work-up of T- and NK-LGL [110,111,112]. 

The 2016 edition of the WHO guidelines established a diagnostic cutoff for LGL as peripheral blood lymphocytosis ranging from 2 to 20 × 10^9^/L persisting for at least 6 months without an identifiable cause [113]. However, it is currently acknowledged that LGL leukemia can manifest even with lower levels of circulating LGL. In these instances, conducting a BM aspirate along with immunohistochemical (IHC) staining can be beneficial in confirming the diagnosis by demonstrating linear arrangements of cytotoxic cells. Subsequently, it becomes crucial to ascertain the presence of clonal LGLs [114]. Immunophenotyping is a crucial aspect of the diagnostic criteria, specifically identifying the T- or NK-cell phenotype. 

LGLs can exhibit polyclonality, suggesting reactive proliferation linked to viral infections such as CMV, HIV, or EBV, particularly in T-LGL [104,115]. Oligoclonal or clonal LGL proliferation may manifest as a post-transplant lymphoproliferative disorder or coexist with other hematological malignancies.

## 6. LGL Leukemia Treatment

There is no standard first-line therapy for LGL leukemia due to a lack of prospective comparative clinical trials. Current therapeutic approaches rely on retrospective or metanalysis of retrospective studies. Protracted immunosuppressive therapy (IST) directed against cytotoxic lymphocytes is the mainstay of treatment in both T- LGL and NK-LGL. Treatment is often continued for a minimum of 4 months prior to assessing the response. Indications for treatment include severe or symptomatic cytopenias and associated autoimmune conditions. Cytopenias include severe neutropenia (ANC < 0.5 × 10^9^/L), moderate neutropenia (ANC > 0.5 × 10^9^/L) with recurrent infection, transfusion-dependent anemia or mixed cytopenias. First-line IST includes low-dose methotrexate (MTX), cyclophosphamide (CTX) and cyclosporine (CsA). Clinical data are readily available for MTX (10 mg/m^2^/weekly), which is the preferred agent in neutropenia and associated rheumatoid arthritis (RA) [116,117], whereas cyclophosphamide (50 to 100 mg/day) is the preferred agent in associated PRCA [118]. Cyclosporin (3–5 mg/kg/day) can be used as first-line or in combination with MTX [24]. The first prospective phase II randomized trial in LGL leukemia investigating first-line therapy using either MTX or CTX is ongoing in France (NCT01976182), and interim results from 96 patients showed a relatively low complete response rate (CRR), (<20%) [119]. 

Response assessment after 4 months of treatment includes hematologic, defined as normalization of blood counts: hemoglobin ≥ 12 g/dL; platelets ≥ 150 × 10^9^/L; ANC ≥ 1.5 × 10^9^/L and lymphocytosis ≤ 4 × 10^9^/L. The reported response rates of these primary IST therapies vary widely between studies. The overall response rate (ORR) for MTX ranges from 38 to 56% [116,120,121,122], with CRR as low as 16% to 21% [43,116]. A prospective phase II study of initial treatment with MTX in LGL leukemia demonstrated an ORR of 37% [123]. MTX is an anti-metabolite most often used as an immunosuppressant in autoimmune diseases; it exerts anti-inflammatory effects by suppressing T-cell activation and enhancing the sensitivity of activated CD-95 T-cells. This effect of MTX is thought to cause the suppression of *JAK/STAT* signaling, and retrospective datasets indicate that the presence of *STAT3* mutation may serve as a predictive factor for MTX responsiveness [124]. 

In comparison, a relatively high ORR (62%) and CRR (32%) has been demonstrated with CTX [116]. CsA has been associated with hematologic response rates ranging from 45% to 56%, and this response has been correlated with the expression of HLA-DR4 [122,125]. Osuji et al. reported ORR on the use of MTX, CsA, and cytotoxic agents in patients with T-cell LGL leukemia and demonstrated an ORR of 78% for CsA [24], while the French Registry showed lower primary response rates in a small number of patients (17%) and ORR of 21% [43]. Refractory disease to IST is not uncommon. In such cases, it is recommended to switch to a different regimen or a combination of regimens. Figure 2 summarizes the treatment algorithm of LGL leukemia based on the available literature. 

### 6.1. Salvage Therapy 

Second-line salvage IST for refractory cases includes sirolimus (rapamycin) and the humanized anti-CD52 monoclonal antibody, alemtuzumab, which has demonstrated an ORR of 56% in a prospective trial [126]. The reported dosage of alemtuzumab is 10 mg 1–2/week subcutaneously [127]. Others include Anti-thymocyte globulins (ATG), which works via the complement-mediated lysis of cytotoxic T cells and rituximab, which is also approved for RA [128]. Fludarabine-based regimens comprising mitoxantrone and dexamethasone have also been utilized with durable remission. At a median follow-up of 71 months, two out of the nine patients were in continuous remission for 6–7 years [129]. Outcomes with stem cell transplantation, either allogeneic hematopoietic or autologous, have also been reported in a multicenter retrospective study [130]. Splenectomy has a reported ORR of 56% and may also be considered in patients with concurrent symptomatic splenomegaly [131]. 

### 6.2. Targeted Therapy 

Other novel targeted therapeutic approaches include *JAK* inhibitors (e.g., ruxolitinib, tofacitinib), IL-6 antagonists and proteosome inhibitors (bortezomib and ixazomib) through the inhibition of the *NF-kB* pathway [132]. Additionally, Siplizumab, an anti-CD2 that targets CD2 expressed on T- and NK cells [133], and BNZ-1, a selective γ-chain cytokine inhibitor, which inhibits IL-2, IL-15 and IL-9 to decrease NK cells, T regulatory and memory cells [134], have been reported. Several other promising therapeutic agents include Slituximab, a chimeric monoclonal antibody that prevents IL-6 binding to its receptor. ABC008 is an anti-KLRG1 antibody that selectively depletes cytotoxic T cells while sparing regulatory T cells and other immune cells. PPMX-T003 is a human anti-transferrin receptor1 (TfR1) antibody that has demonstrated efficacy against hematologic malignancies by inducing apoptosis in tumor cells through the inhibition of TfR1 binding to its ligand, transferrin. KT-333 serves as a selective and potent *STAT3* protein degrader, while DR-01, an anti-CD94 monoclonal antibody, targets cells through antibody-dependent cellular cytotoxicity (ADCC). These innovative agents represent promising avenues for advancing the treatment landscape of LGL leukemia [135,136]. Table 1 shows some of the ongoing clinical trials reported at https://clinicaltrials.gov (accessed on 11 November 2023). 

There is an abundance of evidence suggesting that targeting *STAT3* and *STAT5* signaling could be a promising therapeutic approach against LGL leukemia [137]. Large prospective clinical trials are needed to better deduce therapeutic targets and induce long-lasting remission. 

## 7. Conclusions

LGL leukemia manifests as a chronic lymphoproliferative disease with a wide spectrum of clinical presentations. The management of LGL remains a challenge and is currently incurable. The standard treatment is immunosuppressive agents, albeit with a high relapse rate. Recent progress in understanding the molecular pathways has opened avenues for exploring novel therapeutic approaches, showing promising results. Addressing the urgent unmet need to improve survival requires a concerted effort through well-designed clinical trials guided by a thorough understanding of the disease’s pathogenesis.

## Figures and Tables

**Figure 1 cancers-16-01307-f001:**
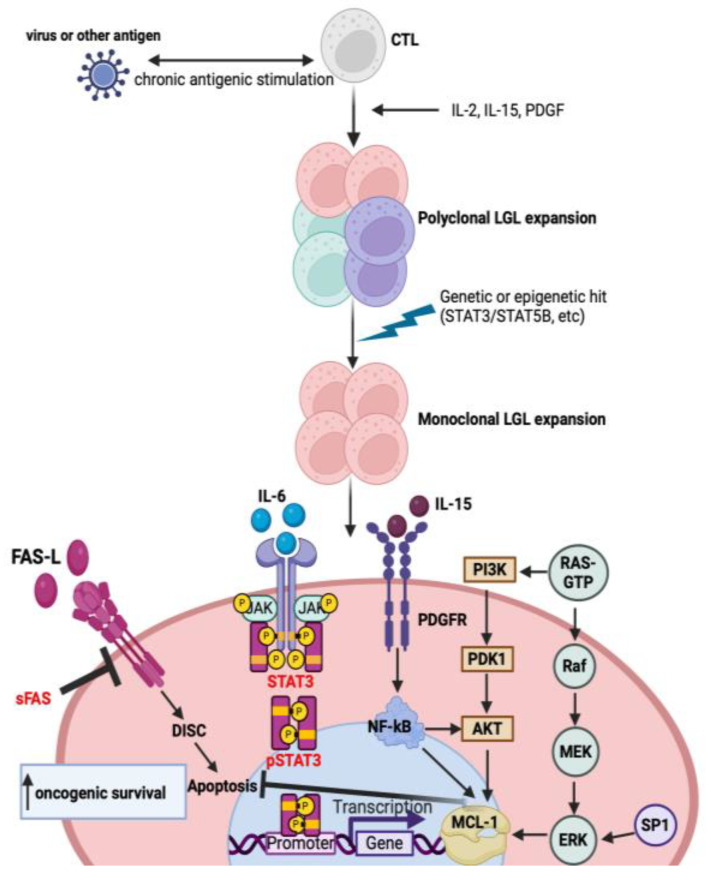
Molecular pathways involved in the pathogenesis of T-LGL leukemia. Chronic antigenic stimulation by a virus leads to polyclonal expansion followed by the dysregulation of cytokines, platelet-derived growth factor (PDGF) and the acquisition of mutations, leading to clonal proliferation promoting leukemic T-LGL cell survival. Constitutive activation of the downstream signaling pathways, such as *JAK-STAT3*, inhibits apoptosis by increasing the transcription of *MCL-1*, an anti-apoptotic protein. Moreover, LGL leukemic cells are resistant to Fas-mediated apoptosis. Soluble Fas (sFas) inhibits Fas-dependent apoptosis by acting as a decoy receptor. Other downstream signaling pathways such as *RAS-MAPK*, *PI3K-AKT*, *SP1-ERK* and *NF-kB* are upregulated in LGL leukemia and modulate the transcription of *MCL-1*, which inhibits apoptosis. These processes lead to the increased transcription of oncogenic driver genes and subsequent malignant cell proliferation and survival.

**Figure 2 cancers-16-01307-f002:**
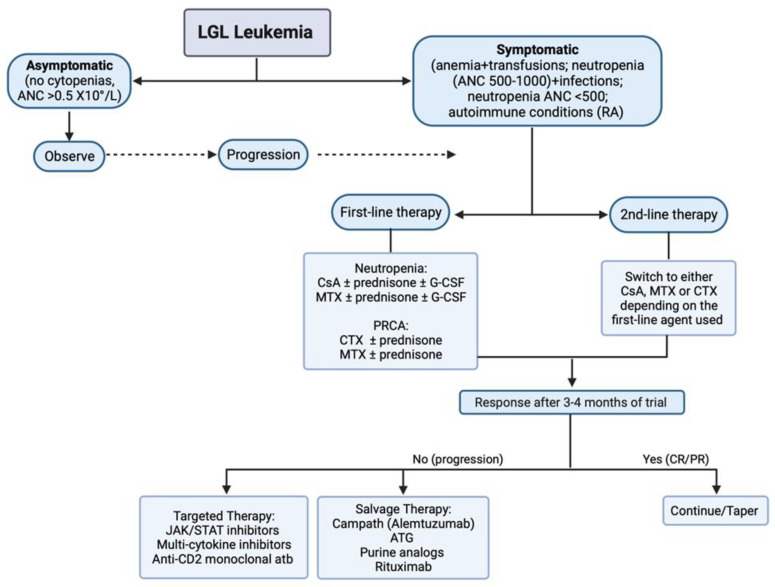
Treatment algorithm in LGL leukemia. Abbreviations: CsA, cyclosporine A; MTX, methotrexate; CTX, cyclophosphamide; ATG, anti-thymocyte globulin; CR, complete response; PR, partial response; atb, antibody.

**Table 1 cancers-16-01307-t001:** Ongoing prospective studies of large granular lymphocytic leukemia.

NCT Number	Study Title	Intervention	Phase	Status
NCT04453345	TPM Regimen (Thalidomide, Prednisone and Methotrexate) in LGLL	Thalidomide + Prednisone + Methotrexate	2 and 3	Recruiting
NCT05316116	Siltuximab in Large Granular Lymphocytic Leukemia (LGLL)	Siltuximab	1	Recruiting
NCT05592015	Ruxolitinib for the Treatment of T-Cell Large Granular Lymphocytic Leukemia	Ruxolitinib	2	Recruiting
NCT05532722	ABC008 in Subjects With T-cell Large Granular Lymphocytic Leukemia (T-LGLL)	ABC008	1 and 2	Recruiting
NCT05141682	Oral Azacitidine for the Treatment of Relapsed or Refractory T-cell Large Granular Lymphocytic Leukemia	Oral Azacitidine	1 and 2	Recruiting
NCT05863234	Safety Evaluation Study for Patients with Aggressive NK-cell Leukemia	PPMX-T003	1 and 2	Recruiting
NCT05225584	Safety, PK, PD, Clinical Activity of KT-333 in Adult Patients with Refractory Lymphoma, Large Granular Lymphocytic Leukemia, Solid Tumors	KT-333	1	Recruiting
NCT05475925	A Study of DR-01 in Subjects with Large Granular Lymphocytic Leukemia or Cytotoxic Lymphomas	DR-01	1 and 2	Recruiting

Abbreviations: PK, pharmacokinetics; PD, pharmacodynamics.

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
