# Peer review of "Large Granular Lymphocytic Leukemia: Clinical Features, Molecular Pathogenesis, Diagnosis and Treatment"

_cancers, 2024, doi:10.3390/cancers16071307_

Round 1

Reviewer 1 Report

Comments and Suggestions for Authors

General: Per WHO-HAEM5, the term LGL within aggressive NK-cell leukemia is not recommended. Considering a fulminant clinical course of aggressive NK-cell leukemia, it would be likely better to focus on T-large granular lymphocyte leukemia (T-LGLL) and NK-large granular lymphocyte leukemia (NK-LGLL) when describing large granular lymphocytic leukemias. Aggressive NK cell leukemia can be discussed when considering differential diagnosis for NK-cell LGL.

1.       Abstract

a.       Page 1 line 21: suggestion … by the persistent clonal expansion… Per WHO-HAEM5 duration of 6 months is not mentioned for T-cell LGL.

b.       Page 1 line 22 and line 23: I would suggest keeping T-large granular lymphocyte leukemia and NK-large granular lymphocyte leukemia but removing aggressive NK cell leukemia and clinical association with aggressive course.

c.       Page 1, line 24-26:

                                                              i.      suggestion …” molecular features, particularly STAT3 and STAT5 mutation in T-large granular lymphocyte leukemia (T-LGLL) and STAT3, TET2, CCL22 mutations in NK-large granular lymphocyte leukemia (NK-LGLL).”

2.       Introduction

a.       Page 2, line 38: suggest removing splenomegaly since it is not common in NK-large granular lymphocyte leukemia.

b.       Page 2, line 50-54,

                                                               i.      Please change mature T-cell malignancies to mature T-cell and NK-cell neoplasms to better reflect WHO-HAEM5 classification and clarify that per WHO-HAEM5 T-large granular lymphocyte leukemia (T-LGLL) and NK-large granular lymphocyte leukemia (NK-LGLL) are within mature T-cell and NK-cell leukemias category. It would be useful also to mention the change in terminology to NK-LGLL leukemia in the WHO-HAEM5

For example: “In general, the WHO-HAEM5 classifies mature T-cell and NK-cell neoplasms into nine distinct categories of which one category is mature T-cell and NK-cell leukemias which includes T-large granular lymphocyte leukemia (T-LGLL) and NK-large granular lymphocyte leukemia (NK-LGLL). Notably, while the term chronic lymphoproliferative disorder of NK cells is kept in ICC, it was replaced with NK-LGLL leukemia in the WHO-HAEM5.”

c.       Page 2, lines 59-62, sentence revision:

                                                               i.      For example, “Conversely, STAT5B mutations which are found in approximately 30% of CD4 cases lack prognostic impact while rare CD8+ have a poor prognosis”.

d.       Page 2, lines 64-66: please clarify the statement. Is the statement about aggressive NK-cell leukemia?

3.       Clinical manifestation of LGL leukemias

a.       Consider having a general description that would refer to both T-large granular lymphocyte leukemia (T-LGLL) and NK-large granular lymphocyte leukemia (NK-LGLL). Consider moving T-LGLL specifics under the T-LGLL section.  

4.       Aggressive NK-cell leukemia

a.       Consider deleting this section.

5.       Pathogenesis

a.       Page 4, line 169: repetition (already mentioned in the introduction)

b.       Please clarify the difference in genetic findings between T-LGL and NK-LGL leukemia.

c.       Page 4, line 178: Please clarify the statement “STAT3 and STAT5 mutations are predictive of poor prognosis.”

d.       Classification

                                                               i.      Consider deleting this section since it is a repetition or removing it from the introduction. Please see the revision suggestions in the introduction section.

e.       Molecular pathway dysregulation

                                                               i.      Page 6, line 239: Suggest changing to LGL cells.

f.        Response to viral antigens:

                                                               i.      HTLV-1 infection? Please clarify if this is only seen with T-LGL leukemia.

g.       Sjogren’s syndrome:

                                                               i.      Please specify association to T-LGL and/or NK LGL leukemia.

h.       Pure red cell aplasia:

                                                               i.      Association with NK LGL leukemia?

6.       Diagnostic criteria for treatment of LGL leukemia

a.       Page 9, line 372 change to immunophenotype

b.       Please consider text revision:

                                                               i.      I would suggest reorganizing text to better reflect steps of diagnostic work-up. When LGL expansion in the blood is observed then immunophenotype or immunohistochemistry is needed to determine lineage: T or NK cells. Based on immunophenotype additional tests are performed. For example, in cases of T-cell expansion testing for T-cell receptor rearrangement while in cases of NK-cell expansion testing for KIR need to be considered…Molecular genetic testing should be considered for both.

                                                             ii.      Page 9, line 391 Consider changing to “in cases of T cell expansion testing for T-cell receptor rearrangement need to be considered”.

Comments on the Quality of English Language

There are some minor grammatical changes for publication, and I would also suggest re-working the text to make it more focused and organized. 

Author Response

Thank you for the opportunity to submit a revised draft of my manuscript titled " Large Granular Lymphocytic Leukemia: Clinical Features, Molecular Pathogenesis, Diagnosis and Treatment". We appreciate the time and effort that you and the reviewers have dedicated to providing your valuable feedback on this manuscript. Below is a summary of the incorporated changes reflecting the suggestions provided by the reviewers. We have highlighted the changes within the manuscript.

Here is a point-by-point response to the reviewer’s comments and concerns.   

Comment 1: Abstract:

  1. Page 1 line 21: suggestion … by the persistent clonal expansion… Per WHO-HAEM5 duration of 6 months is not mentioned for T-cell LGL.
  2. Page 1 line 22 and line 23: I would suggest keeping T-large granular lymphocyte leukemia and NK-large granular lymphocyte leukemia but removing aggressive NK cell leukemia and clinical association with aggressive course.

Response: The suggested changes were incorporated throughout the abstract. Aggressive NK-cell leukemia wasn’t removed for completeness of the sub-types but appreciate the suggestion.

Comment 2: Introduction

  1. Page 2, line 38: suggest removing splenomegaly since it is not common in NK-large granular lymphocyte leukemia.
  2. Page 2, line 50-54, i. Please change mature T-cell malignancies to mature T-cell and NK-cell neoplasms to better reflect WHO-HAEM5 classification and clarify that per WHO-HAEM5 T-large granular lymphocyte leukemia (T-LGLL) and NK-large granular lymphocyte leukemia (NK-LGLL) are within mature T-cell and NK-cell leukemias category. It would be useful also to mention the change in terminology to NK-LGLL leukemia in the WHO-HAEM5

For example: “In general, the WHO-HAEM5 classifies mature T-cell and NK-cell neoplasms into nine distinct categories of which one category is mature T-cell and NK-cell leukemias which includes T-large granular lymphocyte leukemia (T-LGLL) and NK-large granular lymphocyte leukemia (NK-LGLL). Notably, while the term chronic lymphoproliferative disorder of NK cells is kept in ICC, it was replaced with NK-LGLL leukemia in the WHO-HAEM5.”

  1. Page 2, lines 59-62, sentence revision: i.      For example, “Conversely, STAT5B mutations which are found in approximately 30% of CD4 cases lack prognostic impact while rare CD8+ have a poor prognosis”.
  2. Page 2, lines 64-66: please clarify the statement. Is the statement about aggressive NK-cell leukemia?

Response: Greatly appreciate the suggestions. The suggested changes were incorporated throughout the introduction.

Comment 3: Pathogenesis

  1. Page 4, line 169: repetition (already mentioned in the introduction)
  2. Please clarify the difference in genetic findings between T-LGL and NK-LGL leukemia.
  3. Page 4, line 178: Please clarify the statement “STAT3 and STAT5 mutations are predictive of poor prognosis.”
  4. Classification
  5. Consider deleting this section since it is a repetition or removing it from the introduction. Please see the revision suggestions in the introduction section.
  6. Molecular pathway dysregulation
  7. Page 6, line 239: Suggest changing to LGL cells.
  8. Response to viral antigens:
  9. HTLV-1 infection? Please clarify if this is only seen with T-LGL leukemia.
  10. Sjogren’s syndrome:
  11. Please specify association to T-LGL and/or NK LGL leukemia.
  12. Pure red cell aplasia:
  13. Association with NK LGL leukemia?

Response: Thank you for pointing out the errors. The pathogenesis section was extensively revised and the feedback was incorporated accordingly.

Comment 4: Diagnostic criteria for treatment of LGL leukemia

  1. Page 9, line 372 change to immunophenotype
  2. Please consider text revision:
  3. I would suggest reorganizing text to better reflect steps of diagnostic work-up. When LGL expansion in the blood is observed then immunophenotype or immunohistochemistry is needed to determine lineage: T or NK cells. Based on immunophenotype additional tests are performed. For example, in cases of T-cell expansion testing for T-cell receptor rearrangement while in cases of NK-cell expansion testing for KIR need to be considered…Molecular genetic testing should be considered for both.
  4. Page 9, line 391 Consider changing to “in cases of T cell expansion testing for T-cell receptor rearrangement need to be considered”.

Response: The suggested feedback was incorporated into the respective section of the revised manuscript.

In addition to the above comments, all minor comments, spelling and grammatical errors pointed out by the reviewers have been corrected.

Sincerely,

Dr. Fauzia Ullah and the team

Reviewer 2 Report

Comments and Suggestions for Authors

The authors extensively reviewed the clinical pictures, classification, pathogenesis, and treatment of LGL. Although this subject is interesting and informative for hematologists and oncologists, this article has severe problems, for example, insufficient description for molecular pathogenesis of LGL, vague cause and effect of inflammation by viruses or autoimmune disorders, or wrong interpretation of the role of inflammatory cytokines. 

Major comments:

1.       3. Pathogenesis and Figure 1:

This should be the most important part of this article; however, the authors described that chronic antigenic stimulation causes polyclonal LGL expansion that subsequently leads to monoclonal LGL generation through genetic or epigenetic hit toward STAT3/STAT5b in combination with IL2, IL15, and PDGF. It may be reasonable that chronic antigenic stimulation causes polyclonal LGL expansion; however, the authors should explain precise molecular mechanisms how chronic inflammation (IL-2, IL-15, PDGF, etc) genetically or epigenetically hits STAT3/STAT5b and generate mutations of these genes in expanded polyclonal LGL but not other cell types. This issue is crucially important in current scientific paper regarding molecular mechanism of neoplasms.

2.       Regarding autoimmune disorders, cause and effect of these immune disorders on the LGL pathogenesis in unclear in this article.

3.       As viral antigenic stimulation, the authors mentioned only HTLV-1/HTLV-2. How about the role of other virus types. In addition, LGL leukemia is rare in HTLV-1 carrier.

4.       Line 119: It is unlikely that proinflammatory cytokines cause neutropenia.

5.       Line 232: IL-8 causes mild neutrophilia mobilizing neutrophils from marginal pool to blood stream.

6.       Line 234: G-CSF generates neutrophils in the bone marrow and causes neutrophilia in any situations.   

7.       Regarding chronic LGL leukemia, please indicate the following 5 subtypes with surface antigen expression and STAT mutations.

1.     CD3+/CD8+/CD57+: STAT3 mutation

2.     CD3+/CD4+/CD57+: STAT5b mutation

3.     CD3+/CD4+/CD8+/CD57+: STAT3 mutation?

4.     CD3+/CD56+: STAT5b mutation

5.     smCD3-/cyCD3+/CD2dim/CD16+/CD56dim (NK-LGL leukemia): STAT3 mutation

8.       Also, please indicate that CD3+/CD8+/CD57+ LGL leukemia is often associated with autoimmune disorders, while CD3+/CD4+/CD57+ LGL leukemia with monoclonal B lymphocytosis.  

Minor comments:

1.     Please logically make a story of this article avoiding duplication.

2.     Gene name should be written in italic and small character.

3.     Please don’t use capital letter at the head of word unless the word is proper noun.

4.     Please don’t use uncommon abbreviations such as BNZ-1.

5.     Please correct inappropriate English writing.

/

Minor comments:

1.     Please logically make a story of this article avoiding duplication.

2.     Gene name should be written in italic and small character.

3.     Please don’t use capital letter at the head of word unless the word is proper noun.

4.     Please don’t use uncommon abbreviations such as BNZ-1.

5.     Please correct inappropriate English writing.

Major comments:

1.       3. Pathogenesis and Figure 1:

This should be the most important part of this article; however, the authors described that chronic antigenic stimulation causes polyclonal LGL expansion that subsequently leads to monoclonal LGL generation through genetic or epigenetic hit toward STAT3/STAT5b in combination with IL2, IL15, and PDGF. It may be reasonable that chronic antigenic stimulation causes polyclonal LGL expansion; however, the authors should explain precise molecular mechanisms how chronic inflammation (IL-2, IL-15, PDGF, etc) genetically or epigenetically hits STAT3/STAT5b and generate mutations of these genes in expanded polyclonal LGL but not other cell types. This issue is crucially important in current scientific paper regarding molecular mechanism of neoplasms.

2.       Regarding autoimmune disorders, cause and effect of these immune disorders on the LGL pathogenesis in unclear in this article.

3.       As viral antigenic stimulation, the authors mentioned only HTLV-1/HTLV-2. How about the role of other virus types. In addition, LGL leukemia is rare in HTLV-1 carrier.

4.       Line 119: It is unlikely that proinflammatory cytokines cause neutropenia.

5.       Line 232: IL-8 causes mild neutrophilia mobilizing neutrophils from marginal pool to blood stream.

6.       Line 234: G-CSF generates neutrophils in the bone marrow and causes neutrophilia in any situations.   

7.       Regarding chronic LGL leukemia, please indicate the following 5 subtypes with surface antigen expression and STAT mutations.

1.     CD3+/CD8+/CD57+: STAT3 mutation

2.     CD3+/CD4+/CD57+: STAT5b mutation

3.     CD3+/CD4+/CD8+/CD57+: STAT3 mutation?

4.     CD3+/CD56+: STAT5b mutation

5.     smCD3-/cyCD3+/CD2dim/CD16+/CD56dim (NK-LGL leukemia): STAT3 mutation

8.       Also, please indicate that CD3+/CD8+/CD57+ LGL leukemia is often associated with autoimmune disorders, while CD3+/CD4+/CD57+ LGL leukemia with monoclonal B lymphocytosis.  

Minor comments:

1.     Please logically make a story of this article avoiding duplication.

2.     Gene name should be written in italic and small character.

3.     Please don’t use capital letter at the head of word unless the word is proper noun.

4.     Please don’t use uncommon abbreviations such as BNZ-1.

5.     Please correct inappropriate English writing.

The authors extensively reviewed the clinical pictures, classification, pathogenesis, and treatment of LGL. Although this subject is interesting and informative for hematologists and oncologists, this article has severe problems, for example, insufficient description for molecular pathogenesis of LGL, vague cause and effect of inflammation by viruses or autoimmune disorders, or wrong interpretation of the role of inflammatory cytokines. 

Major comments:

1.       3. Pathogenesis and Figure 1:

This should be the most important part of this article; however, the authors described that chronic antigenic stimulation causes polyclonal LGL expansion that subsequently leads to monoclonal LGL generation through genetic or epigenetic hit toward STAT3/STAT5b in combination with IL2, IL15, and PDGF. It may be reasonable that chronic antigenic stimulation causes polyclonal LGL expansion; however, the authors should explain precise molecular mechanisms how chronic inflammation (IL-2, IL-15, PDGF, etc) genetically or epigenetically hits STAT3/STAT5b and generate mutations of these genes in expanded polyclonal LGL but not other cell types. This issue is crucially important in current scientific paper regarding molecular mechanism of neoplasms.

2.       Regarding autoimmune disorders, cause and effect of these immune disorders on the LGL pathogenesis in unclear in this article.

3.       As viral antigenic stimulation, the authors mentioned only HTLV-1/HTLV-2. How about the role of other virus types. In addition, LGL leukemia is rare in HTLV-1 carrier.

4.       Line 119: It is unlikely that proinflammatory cytokines cause neutropenia.

5.       Line 232: IL-8 causes mild neutrophilia mobilizing neutrophils from marginal pool to blood stream.

6.       Line 234: G-CSF generates neutrophils in the bone marrow and causes neutrophilia in any situations.   

7.       Regarding chronic LGL leukemia, please indicate the following 5 subtypes with surface antigen expression and STAT mutations.

1.     CD3+/CD8+/CD57+: STAT3 mutation

2.     CD3+/CD4+/CD57+: STAT5b mutation

3.     CD3+/CD4+/CD8+/CD57+: STAT3 mutation?

4.     CD3+/CD56+: STAT5b mutation

5.     smCD3-/cyCD3+/CD2dim/CD16+/CD56dim (NK-LGL leukemia): STAT3 mutation

8.       Also, please indicate that CD3+/CD8+/CD57+ LGL leukemia is often associated with autoimmune disorders, while CD3+/CD4+/CD57+ LGL leukemia with monoclonal B lymphocytosis.  

Minor comments:

1.     Please logically make a story of this article avoiding duplication.

2.     Gene name should be written in italic and small character.

3.     Please don’t use capital letter at the head of word unless the word is proper noun.

4.     Please don’t use uncommon abbreviations such as BNZ-1.

5.     Please correct inappropriate English writing.

The authors extensively reviewed the clinical pictures, classification, pathogenesis, and treatment of LGL. Although this subject is interesting and informative for hematologists and oncologists, this article has severe problems, for example, insufficient description for molecular pathogenesis of LGL, vague cause and effect of inflammation by viruses or autoimmune disorders, or wrong interpretation of the role of inflammatory cytokines. 

Major comments:

1.       3. Pathogenesis and Figure 1:

This should be the most important part of this article; however, the authors described that chronic antigenic stimulation causes polyclonal LGL expansion that subsequently leads to monoclonal LGL generation through genetic or epigenetic hit toward STAT3/STAT5b in combination with IL2, IL15, and PDGF. It may be reasonable that chronic antigenic stimulation causes polyclonal LGL expansion; however, the authors should explain precise molecular mechanisms how chronic inflammation (IL-2, IL-15, PDGF, etc) genetically or epigenetically hits STAT3/STAT5b and generate mutations of these genes in expanded polyclonal LGL but not other cell types. This issue is crucially important in current scientific paper regarding molecular mechanism of neoplasms.

2.       Regarding autoimmune disorders, cause and effect of these immune disorders on the LGL pathogenesis in unclear in this article.

3.       As viral antigenic stimulation, the authors mentioned only HTLV-1/HTLV-2. How about the role of other virus types. In addition, LGL leukemia is rare in HTLV-1 carrier.

4.       Line 119: It is unlikely that proinflammatory cytokines cause neutropenia.

5.       Line 232: IL-8 causes mild neutrophilia mobilizing neutrophils from marginal pool to blood stream.

6.       Line 234: G-CSF generates neutrophils in the bone marrow and causes neutrophilia in any situations.   

7.       Regarding chronic LGL leukemia, please indicate the following 5 subtypes with surface antigen expression and STAT mutations.

1.     CD3+/CD8+/CD57+: STAT3 mutation

2.     CD3+/CD4+/CD57+: STAT5b mutation

3.     CD3+/CD4+/CD8+/CD57+: STAT3 mutation?

4.     CD3+/CD56+: STAT5b mutation

5.     smCD3-/cyCD3+/CD2dim/CD16+/CD56dim (NK-LGL leukemia): STAT3 mutation

8.       Also, please indicate that CD3+/CD8+/CD57+ LGL leukemia is often associated with autoimmune disorders, while CD3+/CD4+/CD57+ LGL leukemia with monoclonal B lymphocytosis.  

Minor comments:

1.     Please logically make a story of this article avoiding duplication.

2.     Gene name should be written in italic and small character.

3.     Please don’t use capital letter at the head of word unless the word is proper noun.

4.     Please don’t use uncommon abbreviations such as BNZ-1.

5.     Please correct inappropriate English writing.

The authors extensively reviewed the clinical pictures, classification, pathogenesis, and treatment of LGL. Although this subject is interesting and informative for hematologists and oncologists, this article has severe problems, for example, insufficient description for molecular pathogenesis of LGL, vague cause and effect of inflammation by viruses or autoimmune disorders, or wrong interpretation of the role of inflammatory cytokines. 

Major comments:

1.       3. Pathogenesis and Figure 1:

This should be the most important part of this article; however, the authors described that chronic antigenic stimulation causes polyclonal LGL expansion that subsequently leads to monoclonal LGL generation through genetic or epigenetic hit toward STAT3/STAT5b in combination with IL2, IL15, and PDGF. It may be reasonable that chronic antigenic stimulation causes polyclonal LGL expansion; however, the authors should explain precise molecular mechanisms how chronic inflammation (IL-2, IL-15, PDGF, etc) genetically or epigenetically hits STAT3/STAT5b and generate mutations of these genes in expanded polyclonal LGL but not other cell types. This issue is crucially important in current scientific paper regarding molecular mechanism of neoplasms.

2.       Regarding autoimmune disorders, cause and effect of these immune disorders on the LGL pathogenesis in unclear in this article.

3.       As viral antigenic stimulation, the authors mentioned only HTLV-1/HTLV-2. How about the role of other virus types. In addition, LGL leukemia is rare in HTLV-1 carrier.

4.       Line 119: It is unlikely that proinflammatory cytokines cause neutropenia.

5.       Line 232: IL-8 causes mild neutrophilia mobilizing neutrophils from marginal pool to blood stream.

6.       Line 234: G-CSF generates neutrophils in the bone marrow and causes neutrophilia in any situations.   

7.       Regarding chronic LGL leukemia, please indicate the following 5 subtypes with surface antigen expression and STAT mutations.

1.     CD3+/CD8+/CD57+: STAT3 mutation

2.     CD3+/CD4+/CD57+: STAT5b mutation

3.     CD3+/CD4+/CD8+/CD57+: STAT3 mutation?

4.     CD3+/CD56+: STAT5b mutation

5.     smCD3-/cyCD3+/CD2dim/CD16+/CD56dim (NK-LGL leukemia): STAT3 mutation

8.       Also, please indicate that CD3+/CD8+/CD57+ LGL leukemia is often associated with autoimmune disorders, while CD3+/CD4+/CD57+ LGL leukemia with monoclonal B lymphocytosis.  

Minor comments:

1.     Please logically make a story of this article avoiding duplication.

2.     Gene name should be written in italic and small character.

3.     Please don’t use capital letter at the head of word unless the word is proper noun.

4.     Please don’t use uncommon abbreviations such as BNZ-1.

5.     Please correct inappropriate English writing.

The authors extensively reviewed the clinical pictures, classification, pathogenesis, and treatment of LGL. Although this subject is interesting and informative for hematologists and oncologists, this article has severe problems, for example, insufficient description for molecular pathogenesis of LGL, vague cause and effect of inflammation by viruses or autoimmune disorders, or wrong interpretation of the role of inflammatory cytokines. 

Major comments:

1.       3. Pathogenesis and Figure 1:

This should be the most important part of this article; however, the authors described that chronic antigenic stimulation causes polyclonal LGL expansion that subsequently leads to monoclonal LGL generation through genetic or epigenetic hit toward STAT3/STAT5b in combination with IL2, IL15, and PDGF. It may be reasonable that chronic antigenic stimulation causes polyclonal LGL expansion; however, the authors should explain precise molecular mechanisms how chronic inflammation (IL-2, IL-15, PDGF, etc) genetically or epigenetically hits STAT3/STAT5b and generate mutations of these genes in expanded polyclonal LGL but not other cell types. This issue is crucially important in current scientific paper regarding molecular mechanism of neoplasms.

2.       Regarding autoimmune disorders, cause and effect of these immune disorders on the LGL pathogenesis in unclear in this article.

3.       As viral antigenic stimulation, the authors mentioned only HTLV-1/HTLV-2. How about the role of other virus types. In addition, LGL leukemia is rare in HTLV-1 carrier.

4.       Line 119: It is unlikely that proinflammatory cytokines cause neutropenia.

5.       Line 232: IL-8 causes mild neutrophilia mobilizing neutrophils from marginal pool to blood stream.

6.       Line 234: G-CSF generates neutrophils in the bone marrow and causes neutrophilia in any situations.   

7.       Regarding chronic LGL leukemia, please indicate the following 5 subtypes with surface antigen expression and STAT mutations.

1.     CD3+/CD8+/CD57+: STAT3 mutation

2.     CD3+/CD4+/CD57+: STAT5b mutation

3.     CD3+/CD4+/CD8+/CD57+: STAT3 mutation?

4.     CD3+/CD56+: STAT5b mutation

5.     smCD3-/cyCD3+/CD2dim/CD16+/CD56dim (NK-LGL leukemia): STAT3 mutation

8.       Also, please indicate that CD3+/CD8+/CD57+ LGL leukemia is often associated with autoimmune disorders, while CD3+/CD4+/CD57+ LGL leukemia with monoclonal B lymphocytosis.  

Minor comments:

1.     Please logically make a story of this article avoiding duplication.

2.     Gene name should be written in italic and small character.

3.     Please don’t use capital letter at the head of word unless the word is proper noun.

4.     Please don’t use uncommon abbreviations such as BNZ-1.

5.     Please correct inappropriate English writing.

The authors extensively reviewed the clinical pictures, classification, pathogenesis, and treatment of LGL. Although this subject is interesting and informative for hematologists and oncologists, this article has severe problems, for example, insufficient description for molecular pathogenesis of LGL, vague cause and effect of inflammation by viruses or autoimmune disorders, or wrong interpretation of the role of inflammatory cytokines. 

Major comments:

1.       3. Pathogenesis and Figure 1:

This should be the most important part of this article; however, the authors described that chronic antigenic stimulation causes polyclonal LGL expansion that subsequently leads to monoclonal LGL generation through genetic or epigenetic hit toward STAT3/STAT5b in combination with IL2, IL15, and PDGF. It may be reasonable that chronic antigenic stimulation causes polyclonal LGL expansion; however, the authors should explain precise molecular mechanisms how chronic inflammation (IL-2, IL-15, PDGF, etc) genetically or epigenetically hits STAT3/STAT5b and generate mutations of these genes in expanded polyclonal LGL but not other cell types. This issue is crucially important in current scientific paper regarding molecular mechanism of neoplasms.

2.       Regarding autoimmune disorders, cause and effect of these immune disorders on the LGL pathogenesis in unclear in this article.

3.       As viral antigenic stimulation, the authors mentioned only HTLV-1/HTLV-2. How about the role of other virus types. In addition, LGL leukemia is rare in HTLV-1 carrier.

4.       Line 119: It is unlikely that proinflammatory cytokines cause neutropenia.

5.       Line 232: IL-8 causes mild neutrophilia mobilizing neutrophils from marginal pool to blood stream.

6.       Line 234: G-CSF generates neutrophils in the bone marrow and causes neutrophilia in any situations.   

7.       Regarding chronic LGL leukemia, please indicate the following 5 subtypes with surface antigen expression and STAT mutations.

1.     CD3+/CD8+/CD57+: STAT3 mutation

2.     CD3+/CD4+/CD57+: STAT5b mutation

3.     CD3+/CD4+/CD8+/CD57+: STAT3 mutation?

4.     CD3+/CD56+: STAT5b mutation

5.     smCD3-/cyCD3+/CD2dim/CD16+/CD56dim (NK-LGL leukemia): STAT3 mutation

8.       Also, please indicate that CD3+/CD8+/CD57+ LGL leukemia is often associated with autoimmune disorders, while CD3+/CD4+/CD57+ LGL leukemia with monoclonal B lymphocytosis.  

Minor comments:

1.     Please logically make a story of this article avoiding duplication.

2.     Gene name should be written in italic and small character.

3.     Please don’t use capital letter at the head of word unless the word is proper noun.

4.     Please don’t use uncommon abbreviations such as BNZ-1.

5.     Please correct inappropriate English writing.

The authors extensively reviewed the clinical pictures, classification, pathogenesis, and treatment of LGL. Although this subject is interesting and informative for hematologists and oncologists, this article has severe problems, for example, insufficient description for molecular pathogenesis of LGL, vague cause and effect of inflammation by viruses or autoimmune disorders, or wrong interpretation of the role of inflammatory cytokines. 

Major comments:

1.       3. Pathogenesis and Figure 1:

This should be the most important part of this article; however, the authors described that chronic antigenic stimulation causes polyclonal LGL expansion that subsequently leads to monoclonal LGL generation through genetic or epigenetic hit toward STAT3/STAT5b in combination with IL2, IL15, and PDGF. It may be reasonable that chronic antigenic stimulation causes polyclonal LGL expansion; however, the authors should explain precise molecular mechanisms how chronic inflammation (IL-2, IL-15, PDGF, etc) genetically or epigenetically hits STAT3/STAT5b and generate mutations of these genes in expanded polyclonal LGL but not other cell types. This issue is crucially important in current scientific paper regarding molecular mechanism of neoplasms.

2.       Regarding autoimmune disorders, cause and effect of these immune disorders on the LGL pathogenesis in unclear in this article.

3.       As viral antigenic stimulation, the authors mentioned only HTLV-1/HTLV-2. How about the role of other virus types. In addition, LGL leukemia is rare in HTLV-1 carrier.

4.       Line 119: It is unlikely that proinflammatory cytokines cause neutropenia.

5.       Line 232: IL-8 causes mild neutrophilia mobilizing neutrophils from marginal pool to blood stream.

6.       Line 234: G-CSF generates neutrophils in the bone marrow and causes neutrophilia in any situations.   

7.       Regarding chronic LGL leukemia, please indicate the following 5 subtypes with surface antigen expression and STAT mutations.

1.     CD3+/CD8+/CD57+: STAT3 mutation

2.     CD3+/CD4+/CD57+: STAT5b mutation

3.     CD3+/CD4+/CD8+/CD57+: STAT3 mutation?

4.     CD3+/CD56+: STAT5b mutation

5.     smCD3-/cyCD3+/CD2dim/CD16+/CD56dim (NK-LGL leukemia): STAT3 mutation

8.       Also, please indicate that CD3+/CD8+/CD57+ LGL leukemia is often associated with autoimmune disorders, while CD3+/CD4+/CD57+ LGL leukemia with monoclonal B lymphocytosis.  

Minor comments:

1.     Please logically make a story of this article avoiding duplication.

2.     Gene name should be written in italic and small character.

3.     Please don’t use capital letter at the head of word unless the word is proper noun.

4.     Please don’t use uncommon abbreviations such as BNZ-1.

5.     Please correct inappropriate English writing.

The authors extensively reviewed the clinical pictures, classification, pathogenesis, and treatment of LGL. Although this subject is interesting and informative for hematologists and oncologists, this article has severe problems, for example, insufficient description for molecular pathogenesis of LGL, vague cause and effect of inflammation by viruses or autoimmune disorders, or wrong interpretation of the role of inflammatory cytokines. 

Major comments:

1.       3. Pathogenesis and Figure 1:

This should be the most important part of this article; however, the authors described that chronic antigenic stimulation causes polyclonal LGL expansion that subsequently leads to monoclonal LGL generation through genetic or epigenetic hit toward STAT3/STAT5b in combination with IL2, IL15, and PDGF. It may be reasonable that chronic antigenic stimulation causes polyclonal LGL expansion; however, the authors should explain precise molecular mechanisms how chronic inflammation (IL-2, IL-15, PDGF, etc) genetically or epigenetically hits STAT3/STAT5b and generate mutations of these genes in expanded polyclonal LGL but not other cell types. This issue is crucially important in current scientific paper regarding molecular mechanism of neoplasms.

2.       Regarding autoimmune disorders, cause and effect of these immune disorders on the LGL pathogenesis in unclear in this article.

3.       As viral antigenic stimulation, the authors mentioned only HTLV-1/HTLV-2. How about the role of other virus types. In addition, LGL leukemia is rare in HTLV-1 carrier.

4.       Line 119: It is unlikely that proinflammatory cytokines cause neutropenia.

5.       Line 232: IL-8 causes mild neutrophilia mobilizing neutrophils from marginal pool to blood stream.

6.       Line 234: G-CSF generates neutrophils in the bone marrow and causes neutrophilia in any situations.   

7.       Regarding chronic LGL leukemia, please indicate the following 5 subtypes with surface antigen expression and STAT mutations.

1.     CD3+/CD8+/CD57+: STAT3 mutation

2.     CD3+/CD4+/CD57+: STAT5b mutation

3.     CD3+/CD4+/CD8+/CD57+: STAT3 mutation?

4.     CD3+/CD56+: STAT5b mutation

5.     smCD3-/cyCD3+/CD2dim/CD16+/CD56dim (NK-LGL leukemia): STAT3 mutation

8.       Also, please indicate that CD3+/CD8+/CD57+ LGL leukemia is often associated with autoimmune disorders, while CD3+/CD4+/CD57+ LGL leukemia with monoclonal B lymphocytosis.  

Minor comments:

1.     Please logically make a story of this article avoiding duplication.

2.     Gene name should be written in italic and small character.

3.     Please don’t use capital letter at the head of word unless the word is proper noun.

4.     Please don’t use uncommon abbreviations such as BNZ-1.

5.     Please correct inappropriate English writing.

The authors extensively reviewed the clinical pictures, classification, pathogenesis, and treatment of LGL. Although this subject is interesting and informative for hematologists and oncologists, this article has severe problems, for example, insufficient description for molecular pathogenesis of LGL, vague cause and effect of inflammation by viruses or autoimmune disorders, or wrong interpretation of the role of inflammatory cytokines. 

Major comments:

1.       3. Pathogenesis and Figure 1:

This should be the most important part of this article; however, the authors described that chronic antigenic stimulation causes polyclonal LGL expansion that subsequently leads to monoclonal LGL generation through genetic or epigenetic hit toward STAT3/STAT5b in combination with IL2, IL15, and PDGF. It may be reasonable that chronic antigenic stimulation causes polyclonal LGL expansion; however, the authors should explain precise molecular mechanisms how chronic inflammation (IL-2, IL-15, PDGF, etc) genetically or epigenetically hits STAT3/STAT5b and generate mutations of these genes in expanded polyclonal LGL but not other cell types. This issue is crucially important in current scientific paper regarding molecular mechanism of neoplasms.

2.       Regarding autoimmune disorders, cause and effect of these immune disorders on the LGL pathogenesis in unclear in this article.

3.       As viral antigenic stimulation, the authors mentioned only HTLV-1/HTLV-2. How about the role of other virus types. In addition, LGL leukemia is rare in HTLV-1 carrier.

4.       Line 119: It is unlikely that proinflammatory cytokines cause neutropenia.

5.       Line 232: IL-8 causes mild neutrophilia mobilizing neutrophils from marginal pool to blood stream.

6.       Line 234: G-CSF generates neutrophils in the bone marrow and causes neutrophilia in any situations.   

7.       Regarding chronic LGL leukemia, please indicate the following 5 subtypes with surface antigen expression and STAT mutations.

1.     CD3+/CD8+/CD57+: STAT3 mutation

2.     CD3+/CD4+/CD57+: STAT5b mutation

3.     CD3+/CD4+/CD8+/CD57+: STAT3 mutation?

4.     CD3+/CD56+: STAT5b mutation

5.     smCD3-/cyCD3+/CD2dim/CD16+/CD56dim (NK-LGL leukemia): STAT3 mutation

8.       Also, please indicate that CD3+/CD8+/CD57+ LGL leukemia is often associated with autoimmune disorders, while CD3+/CD4+/CD57+ LGL leukemia with monoclonal B lymphocytosis.  

Minor comments:

1.     Please logically make a story of this article avoiding duplication.

2.     Gene name should be written in italic and small character.

3.     Please don’t use capital letter at the head of word unless the word is proper noun.

4.     Please don’t use uncommon abbreviations such as BNZ-1.

5.     Please correct inappropriate English writing.

Comments on the Quality of English Language

Please refer minor comments.

Author Response

Thank you for the opportunity to submit a revised draft of my manuscript titled " Large Granular Lymphocytic Leukemia: Clinical Features, Molecular Pathogenesis, Diagnosis and Treatment". We appreciate the time and effort that you have dedicated to providing your valuable feedback on this manuscript. Below is a summary of the incorporated changes reflecting the suggestions provided by you. We have highlighted the changes within the manuscript.

Here is a point-by-point response to the reviewer’s comments and concerns.   

Comment 1: Pathogenesis and Figure 1: This should be the most important part of this article; however, the authors described that chronic antigenic stimulation causes polyclonal LGL expansion that subsequently leads to monoclonal LGL generation through genetic or epigenetic hit toward STAT3/STAT5b in combination with IL2, IL15, and PDGF. It may be reasonable that chronic antigenic stimulation causes polyclonal LGL expansion; however, the authors should explain precise molecular mechanisms how chronic inflammation (IL-2, IL-15, PDGF, etc) genetically or epigenetically hits STAT3/STAT5b and generate mutations of these genes in expanded polyclonal LGL but not other cell types. This issue is crucially important in current scientific paper regarding molecular mechanism of neoplasms.

Response: Thank you for pointing this out. The pathogenesis section was extensively revised and figure 1 has been edited. Please refer to the corresponding section in the manuscript.   

Comment 2:  Regarding autoimmune disorders, cause and effect of these immune disorders on the LGL pathogenesis in unclear in this article.

Response: This section was revised accordingly.

Comment 3:  As viral antigenic stimulation, the authors mentioned only HTLV-1/HTLV-2. How about the role of other virus types. In addition, LGL leukemia is rare in HTLV-1 carrier.

Response: This section was revised accordingly.

Comment 7: Regarding chronic LGL leukemia, please indicate the following 5 subtypes with surface antigen expression and STAT mutations. 

   CD3+/CD8+/CD57+: STAT3 mutation

CD3+/CD4+/CD57+: STAT5b mutation

CD3+/CD4+/CD8+/CD57+: STAT3 mutation?

CD3+/CD56+: STAT5b mutation

smCD3-/cyCD3+/CD2dim/CD16+/CD56dim (NK-LGL leukemia): STAT3 mutation

Response: Thank you for this very interesting comment. We are open to incorporating these changes but would like a reference(s). Otherwise, we will leave it as is since this manuscript is not solely about the pathogenesis or surface markers.   

In addition to the above comments, all minor comments, spelling and grammatical errors pointed out by the reviewers have been corrected.

Sincerely,

Dr. Fauzia Ullah and the team

Round 2

Reviewer 1 Report

Comments and Suggestions for Authors

Reviewer 1 commentary, 2nd round

ABSTRACT AND TEXT

Dear authors, please note that making a diagnosis is based on current guidelines and criteria which are determined per WHO-HAEM5 and/or ICC (also stated in this manuscript). As I mentioned in a prior review per WHO-HAEM5 the term LGL within aggressive NK-cell leukemia is not recommended (please see WHO-HAEM5 under the chapter “Aggressive NK-cell leukemia” and look for not recommended terminology). Thus, aggressive NK cell leukemia is not a sub-type of LGL.

Page 1 line 23: Please delete the sentence in abstract “The three sub-types include chronic T-LGL, NK-LGL and aggressive T-LGL or NK cell leukemia.” Please see the explanation in the prior paragraph. Please note that when talking about LGL, then T-large granular lymphocytic leukemia and NK-large granular lymphocytic leukemia (per ICC: Chronic lymphoproliferative disorder of NK cells) are usually considered. I would not necessarily categorize them as subtypes, since that is not how they are listed per WHO-HAEM5 and/or ICC.

1. INTRODUCTION

Page 2, lines 53-55: I would suggest further clarification of sub-classification “T-cell malignancies and NK-cell neoplasms” per WHO-HAEM5.

Suggestion: In general, the WHO-HAEM5 classifies mature T-cell malignancies and NK-cell neoplasms into nine distinct categories. One of the nine categories is mature T-cell and NK-cell leukemias which includes six entities representative of T-and NK-cell proliferations that primarily present as leukemia disease and include the following three: T-cell large granular lymphocyte (T-LGL) leukemia, NK-cell large granular lymphocyte (NK-LGL) leukemia, and aggressive NK-cell leukemia (ANKL).

Page 2, lines 68-70: Please revise.

Line 60-62, STAT3 is associated with poor survival in T-LGL. Thus, lines 68-70 statement that the indolent subtype commonly features STAT3 mutations is contraindicatory. I would suggest to delete line 68-70.

Page 2, line 74: Please delete the sentence “However, individuals with ANKL have a grim prognosis due to refractoriness to chemotherapy.” This statement does not add any relevance to the paragraph.

2. CLINICAL MANIFESTATION OF LGL LEUKEMIA (pages 2-4, lines 83-169)

Page 2-4, lines 83-169: As previously mentioned, please revise this paragraph entirely and describe general findings for LGL (specifically T-LGL and NK-LGL), then clearly state clinical manifestation for T-LGL vs. NK-LGL. Please avoid repetition and try to be more focused. Please also see comment below for 4. ASSOCIATED AUTOIMMUNE AND BONE MARROW FAILURE DISORDERS.

Page 3, lines 88-91: Please delete “however, the prognosis of aggressive NK-cell leukemia is very poor, with an OS as low as 58 days [13,14].” Aggressive NK-cell leukemia is NOT a subtype of LGL leukemia as previously discussed.

Page 3 lines 92-113: Sections describing the clinical manifestation of T-LGL should be moved under the T-LGL leukemia section.

Page 3, lines 114-115

The title” Subtype-specific manifestation” can be deleted. As previously mentioned, they are not categorized as subtypes per WHO-HAEM5 and/or ICC.

Page 4, lines 164-169

Please delete the section “Aggressive NK-cell leukemia” as previously stated aggressive NK-cell leukemia is NOT a subtype of LGL leukemia. As mentioned in a prior review aggressive NK-cell leukemia could be mentioned in discussion when considering differential diagnosis.

3. PATHOGENESIS

Page 8, line 328: Please clarify how EBV is associated with T-LGL and/or NK-LGL, give references.

Page 8, line 335: Please clarify what type of LGL is associated with Hepatitis C.

4. ASSOCIATED AUTOIMMUNE AND BONE MARROW FAILURE DISORDERS, pages 8-10

Page 8-10: Repetition (if autoimmune disorders are to be discussed in this section, then it should be moved from “Clinical manifestation of LGL leukemia (pages 2-4, lines 83-169) section”. Please specify if it is T-LGL and/or NK-LGL that is associated with autoimmune and bone marrow failure disorders. As mentioned in the 1st review, it would be interesting to know about the association of autoimmune and bone marrow failure disorders with NK-LGL.

5. DIAGNOSTIC CRITERIA FOR TREATMENT OF LGL LEUKEMIA

Page 10, line 421; REVISE text: T-cell receptor (TCR) rearrangement would be ONLY considered for T cell expansion, thus, for T-LGL. I would suggest using a broader term, such as “molecular analysis”. Suggestion: “The diagnosis of LGL leukemia relies on key criteria including clinical presentation, cellular morphology, immunophenotype, and molecular analysis.”

Page 10, lines 422 – 427 and page 11, lines 440-452 should be merged to have better flow. For better clarity and organization please suggestion below.

Page 10, lines 422-425; suggestion “Examination of the peripheral blood morphology and evaluation of lymphocyte cell counts, immunophenotyping by flow cytometry utilizing markers such as CD3, CD5, CD4, CD8, CD16, CD27, CD28, CD45, CD56, CD57, CD62, CD94 should be conducted. Additionally, evaluation of TCR beta chain constant region (TRBC1) or TCRVβ, and killer cell immunoglobulin-like receptor (KIR; CD158) expression, can be assessed by flow cytometry [33,104,105].” Consider incorporating lines: 448-452…In case of suspected T-cell LGL, testing for TCR rearrangement is performed. Molecular genetics testing, including evaluation for STAT3/STAT5 alterations should be included in diagnostic work-up of T and NK-LGL.”

Page 10, line 426 note “rearrangement” is not the correct terminology.

Page 11, lines 437-438 Please be specific if it is T-LGL and/or NK-LGL that is linked to viral infections such as CMV, HIV, or EBV.

Page 11, lines 445-446 is a repetition and can be deleted.

Page 11, line 447 should be moved (these are markers used by flow analysis, please see suggestion above).  

Page 11, lines 448-449 should be moved to a section where flow cytometry is discussed (please see suggestion above).

Page 11, lines 449-452 (please see suggestion above).

7. CONCLUSION

Page 13, line 532-533; Please revise sentence and remove aggressive presentation. Aggressive NK-cell leukemia is NOT a subtype of LGL leukemia as previously discussed.

Comments on the Quality of English Language

Minor editing of English language required.

Author Response

Dear reviewer, 

Thank you for the opportunity to submit a revised draft of my manuscript titled " Large Granular Lymphocytic Leukemia: Clinical Features, Molecular Pathogenesis, Diagnosis and Treatment". We appreciate the time and effort that you have dedicated to providing your valuable feedback on this manuscript. Below is a summary of the incorporated changes reflecting the suggestions provided by you. We have highlighted the changes within the manuscript.

Comment 1: Abstract:

  1. Page 1 line 23: Please delete the sentence in abstract “The three sub-types include chronic T-LGL, NK-LGL and aggressive T-LGL or NK cell leukemia.” Please see the explanation in the prior paragraph. Please note that when talking about LGL, then T-large granular lymphocytic leukemia and NK-large granular lymphocytic leukemia (per ICC:Chronic lymphoproliferative disorder of NK cells) are usually considered. I would not necessarily categorize them as subtypes, since that is not how they are listed per WHO-HAEM5 and/or ICC.

Response: Your constructive feedback is greatly appreciated. The sentence was deleted. Please see lines 22-24.  

Comment 2: Introduction

  1. Page 2, lines 53-55: I would suggest further clarification of sub-classification “T-cell malignancies and NK-cell neoplasms” per WHO-HAEM5. Suggestion: In general, the WHO-HAEM5 classifies mature T-cell malignancies and NK-cell neoplasms into nine distinct categories. One of the nine categories is mature T-cell and NK-cell leukemias which includes six entities representative of T-and NK-cell proliferations that primarily present as leukemia disease and include the following three: T-cell large granular lymphocyte (T-LGL) leukemia, NK-cell large granular lymphocyte (NK-LGL) leukemia, and aggressive NK-cell leukemia (ANKL).Page 2, lines 68-70: Please revise.
  2. Line 60-62, STAT3 is associated with poor survival in T-LGL. Thus, lines 68-70 statement that the indolent subtype commonly features STAT3 mutations is contraindicatory. I would suggest to delete line 68-70.
  3. Page 2, line 74: Please delete the sentence “However, individuals with ANKL have a grim prognosis due to refractoriness to chemotherapy.” This statement does not add any relevance to the paragraph.

Response: Greatly appreciate the suggestions. The suggested changes were incorporated throughout the introduction.

  1. Please see lines 53-59
  2. Please see lines 72-74
  3. Please see lines 78-79

Comment 3: CLINICAL MANIFESTATION OF LGL LEUKEMIA (pages 2-4, lines 83-169)

  1. Page 2-4, lines 83-169: As previously mentioned, please revise this paragraph entirely and describe general findings for LGL (specifically T-LGL and NK-LGL), then clearly state clinical manifestation for T-LGL vs. NK-LGL. Please avoid repetition and try to be more focused. Please also see comment below for 4. ASSOCIATED AUTOIMMUNE AND BONE MARROW FAILURE DISORDERS.
  2. Page 3, lines 88-91: Please delete “however, the prognosis of aggressive NK-cell leukemia is very poor, with an OS as low as 58 days [13,14].” Aggressive NK-cell leukemia is NOT a subtype of LGL leukemia as previously discussed.
  3. Page 3 lines 92-113: Sections describing the clinical manifestation of T-LGL should be moved under the T-LGL leukemia section.
  4. Page 3, lines 114-115. The title” Subtype-specific manifestation” can be deleted. As previously mentioned, they are not categorized as subtypes per WHO-HAEM5 and/or ICC.
  5. Page 4, lines 164-169. Please delete the section “Aggressive NK-cell leukemia” as previously stated aggressive NK-cell leukemia is NOT a subtype of LGL leukemia. As mentioned in a prior review aggressive NK-cell leukemia could be mentioned in discussion when considering differential diagnosis.

Response: Thank you again for your feedback.

  1. Please see lines 88-102.
  2. Please see lines 98 -100.
  3. The entire section was moved and T-LGL leukemia section was revised. Please see lines 129-181.
  4. This section was removed. Please see lines 125-127.
  5. This section was deleted. Please see lines 220-225. But added to NK-LGL section as a rare proliferation rather than as a subtype. Please see 214-219.

Comment 4: Pathogenesis

  1. Page 8, line 328: Please clarify how EBV is associated with T-LGL and/or NK-LGL, give references.
  2. Page 8, line 335: Please clarify what type of LGL is associated with Hepatitis C.

Response:

  1. Please see lines 394-395.
  2. Please see lines 400-402.

Comment 5: ASSOCIATED AUTOIMMUNE AND BONE MARROW FAILURE DISORDERS, pages 8-10. Page 8-10: Repetition (if autoimmune disorders are to be discussed in this section, then it should be moved from “Clinical manifestation of LGL leukemia (pages 2-4, lines 83-169) section”. Please specify if it is T-LGL and/or NK-LGL that is associated with autoimmune and bone marrow failure disorders. As mentioned in the 1st review, it would be interesting to know about the association of autoimmune and bone marrow failure disorders with NK-LGL.

Response: The entire section was revised. Most of the diseases are associated with T-LGL and this was added to each disorder.

Comment 6: DIAGNOSTIC CRITERIA FOR TREATMENT OF LGL LEUKEMIA

  1. Page 10, line 421; REVISE text: T-cell receptor (TCR) rearrangement would be ONLY considered for T cell expansion, thus, for T-LGL. I would suggest using a broader term, such as “molecular analysis”. Suggestion: “The diagnosis of LGL leukemia relies on key criteria including clinical presentation, cellular morphology, immunophenotype, and molecular analysis.”
  2. Page 10, lines 422 – 427 and page 11, lines 440-452 should be merged to have better flow. For better clarity and organization please suggestion below. Page 10, lines 422-425; suggestion “Examination of the peripheral bloodmorphology and evaluation of lymphocyte cell counts, immunophenotyping by flow cytometry utilizing markers such as CD3, CD5, CD4, CD8, CD16, CD27, CD28, CD45, CD56, CD57, CD62, CD94 should be conducted. Additionally, evaluation of TCR beta chain constant region (TRBC1) or TCRVβ, and killer cell immunoglobulin-like receptor (KIR; CD158) expression, can be assessed by flow cytometry [33,104,105].” Consider incorporating lines: 448-452…In case of suspected T-cell LGL, testing for TCR rearrangement is performed. Molecular genetics testing, including evaluation for STAT3/STAT5 alterations should be included in diagnostic work-up of T and NK-LGL.”
  3. Page 10, line 426 note “rearrangement” is not the correct terminology.
  4. Page 11, lines 437-438 Please be specific if it is T-LGL and/or NK-LGL that is linked to viral infections such as CMV, HIV, or EBV.
  5. Page 11, lines 445-446 is a repetition and can be deleted.
  6. Page 11, line 447 should be moved (these are markers used by flow analysis, please see suggestion above).
  7. Page 11, lines 448-449 should be moved to a section where flow cytometry is discussed (please see suggestion above).
  8. Page 11, lines 449-452 (please see suggestion above).

Response: Greatly appreciate the suggestions.

  1. Please see lines 502-505
  2. Please see lines 505-515
  3. This was deleted
  4. T-LGL was added to this section
  5. This was deleted
  6. This was deleted due to repetition. Please see lines 535-540
  7. This was corrected
  8. Corrections were made as suggested

Comment 7: CONCLUSION. Page 13, line 532-533; Please revise sentence and remove aggressive presentation. Aggressive NK-cell leukemia is NOT a subtype of LGL leukemia as previously discussed. 

Response: The sentence was revised. Please see lines 622-623.

Sincerely,

Dr. Ullah and the team 

Reviewer 2 Report

Comments and Suggestions for Authors

The authors have reasonably revised the former manuscript in some points; however, the molecular pathogenesis of LGL leukemia is still insufficient. Also, description of clinical features is unsatisfactory.

Major comments:

1.        In the section of 3. Pathogenesis, molecular mechanism regarding the generation of STAT3/STAT5b mutations in LGL is incompletely reviewed. The authors’ statement is rather the effect of STAT gene mutations on the proliferation of LGL leukemia but not cause of the mutations. While, the authors cited Ref. No 54 in page 7, line 251 to 252. This study is very important in this article in terms of molecular mechanism of the development of LGL leukemia. Chromosomal instability and DNA hypermethylation caused by chronic IL-15 exposure may lead to the generation of STAT3/STAT5b mutations. Therefore, the authors should mention about this study (Ref. No 54) in the section of Pathogenesis and review the molecular mechanism of the gene mutation development.

2.       As the molecular mechanism of the gene mutation, the authors extensively described IL-15 signaling pathways in page 7; however, this paragraph does not point molecular pathogenesis of LGL leukemia: Therefore, this paragraph should be shortened avoiding the duplication IL-15 exposure-caused DNA alteration in the section of Pathogenesis.

3.       As clinical feature section of this article, description 5 subtypes of chronic LGL leukemia along with surface antigen expression and STAT mutations is still important. Please read the following paper; J Clin Exp Hematop 2019; 59: 202-206. The authors will find respective references for the 5 subtypes.

4.       Also, please indicate that CD3+/CD8+/CD57+ LGL leukemia is often associated with autoimmune disorders, while CD3+/CD4+/CD57+ LGL leukemia with monoclonal B lymphocytosis. The authors find the references regarding these clinical features in the above article.

5.       Again, gene name should be written in italic.

Comments on the Quality of English Language

Please refer to my commemts.

Author Response

Dear Reviewer, 

Thank you for the opportunity to submit a revised draft of my manuscript titled " Large Granular Lymphocytic Leukemia: Clinical Features, Molecular Pathogenesis, Diagnosis and Treatment". We appreciate the time and effort that you have dedicated to providing your valuable feedback on this manuscript. Below is a summary of the incorporated changes reflecting the suggestions provided by you. We have highlighted the changes within the manuscript.

Comment 1: In the section of 3. Pathogenesis, molecular mechanism regarding the generation of STAT3/STAT5b mutations in LGL is incompletely reviewed. The authors’ statement is rather the effect of STAT gene mutations on the proliferation of LGL leukemia but not cause of the mutations. While, the authors cited Ref. No 54 in page 7, line 251 to 252. This study is very important in this article in terms of molecular mechanism of the development of LGL leukemia. Chromosomal instability and DNA hypermethylation caused by chronic IL-15 exposure may lead to the generation of STAT3/STAT5b mutations. Therefore, the authors should mention about this study (Ref. No 54) in the section of Pathogenesis and review the molecular mechanism of the gene mutation development.

Response: Greatly appreciate your feedback and for pointing out the reference. This section was revised. Please see lines 229-243. 

Comment 2:  As the molecular mechanism of the gene mutation, the authors extensively described IL-15 signaling pathways in page 7; however, this paragraph does not point molecular pathogenesis of LGL leukemia: Therefore, this paragraph should be shortened avoiding the duplication IL-15 exposure-caused DNA alteration in the section of Pathogenesis.

Response: This paragraph was shortened. Please lines 298-314. Repetition was corrected.

Comment 3:  As clinical feature section of this article, description 5 subtypes of chronic LGL leukemia along with surface antigen expression and STAT mutations is still important. Please read the following paper; J Clin Exp Hematop 2019; 59: 202-206. The authors will find respective references for the 5 subtypes.

Response: Greatly appreciate for providing the reference. This was incorporated into lines 356-360.

Comment 4:  Also, please indicate that CD3+/CD8+/CD57+ LGL leukemia is often associated with autoimmune disorders, while CD3+/CD4+/CD57+ LGL leukemia with monoclonal B lymphocytosis. The authors find the references regarding these clinical features in the above article.

Response: Thank you for your suggestions. This was incorporated into lines 407-409.

Comment 5:  Again, gene name should be written in italic.   

Response: All gene names throughout the manuscript were changed to italic. If anything is still missing, any feedback is greatly appreciated.

Sincerely, 

Dr. Ullah and the team 

Round 3

Reviewer 1 Report

Comments and Suggestions for Authors

Reviewer 1 commentary: 

2.1 T-LGL leukemia 

a. Page 4, line 163: abbreviation RA used for the first time in text... please insert rheumatoid arthritis (RA) 

2.2 NK-LGL leukemia 

a. Page 5, line 214: recommend inserting “the differential diagnosis of NK-LGL leukemia also include aggressive NK-cell leukemia” before starting sentence: Aggressive 214 NK-LGL leukemia is a rare LGL proliferation... 

3.23. Molecular pathway dysregulation 

a. Page 9, lines 356-359: please check accuracy (also see the statement made about immunophenotype on page 2, lines 43-44) 

- CD3+/CD8+/CD57+, CD3+/CD4+/CD8dim/CD57+ points to T-LGL  

- What cyc stands for when mentioning smCD3-/cycCD3+/CD2dim/CD16+/CD56dim  

- I would suggest deleting this sentence since the immunophenotype stated is for identification of either T-LGL or NK-LGL. Also, prior sentences in this chapter gave more detailed information about STAT3 and STAT5b mutations in T-LGL and NK-LGL.  

4.2. LGL and bone marrow failure disorders 

a. Page 10, line 407: abbreviation PRCA used for the first time in text... please insert pure red cell aplasia (PRCA) 

5. Diagnostic criteria for treatment of LGL leukemia  

a. Page 12, line 513: For consistency please change T cell LGL to T-LGL

6. LGL leukemia treatment 

a. page 13, lines 545-546: Please change T-cell LGL and NK-cell leukemia to T-LGL and NK-LGL 

Author Response

Dear reviewer,

Thank you for the opportunity to submit a revised draft of my manuscript titled " Large Granular Lymphocytic Leukemia: Clinical Features, Molecular Pathogenesis, Diagnosis and Treatment". We appreciate the time and effort that you have dedicated to providing your valuable feedback on this manuscript. Below is a summary of the incorporated changes reflecting the suggestions provided by you. We have highlighted the changes within the manuscript.

Here is a point-by-point response to the reviewer’s comments and concerns.   

Comment 1: T-LGL leukemia: Page 4, line 163, abbreviation RA used for the first time in text... please insert rheumatoid arthritis (RA) 

Response: Your constructive and detailed feedback is greatly appreciated. This was changed. Please see line 163

Comment 2: NK-LGL leukemia: Page 5, line 214: recommend inserting “thedifferential diagnosisof NK-LGL leukemia also include aggressive NK-cell leukemia” before starting sentence: Aggressive 214 NK-LGL leukemia is a rare LGL proliferation... 

Response: Your suggestion was incorporated. Please see lines 214-215

Comment 3: Molecular pathway dysregulation: Page 9, lines356-359: please check accuracy (also see the statement made about immunophenotype on page 2, lines 43-44) 

  • CD3+/CD8+/CD57+, CD3+/CD4+/CD8dim/CD57+ points to T-LGL
  • What cyc stands for when mentioning smCD3-/cycCD3+/CD2dim/CD16+/CD56dim 
  • I would suggest deleting this sentence since the immunophenotype stated is for identification of either T-LGL or NK-LGL. Also, prior sentences in this chapter gave more detailed information about STAT3 and STAT5b mutations in T-LGL and NK-LGL.  

Response: Thank you for the detailed feedback and we have greatly benefited from your expertise. Lines 43-44 were updated. Lines 356-359, references were checked for accuracy. As for the phenotypes, the other reviewer suggested that these specific ones be incorporated in the manuscript, however, this “smCD3-/cycCD3+/CD2dim/CD16+/CD56dim” was deleted due to lack of mention in the reference.

Comments 4, 5, 6: LGL and bone marrow failure disorders: Page 10, line 407:abbreviation PRCA used for the first time in text... please insert pure red cell aplasia (PRCA) 

Diagnostic criteria for treatment of LGL leukemia: Page 12, line513: For consistency please change T cell LGL to T-LGL 

LGL leukemia treatment: page 13, lines 545-546:Please change T-cell LGL and NK-cell leukemia to T-LGL and NK-LGL 

Response: Thank you for checking for consistency. Lines 407, 513, 545-546 were updated as suggested.

Sincerely,

Dr. Ullah and the team

Reviewer 2 Report

Comments and Suggestions for Authors

The manuscript has been reasonably revised in terms of molecular pthogenesis and climical features of  LGL leukemias.

Comments on the Quality of English Language

line 339: Epidermal Growth Fcactor (EGF) should be as epidermal growth factor (EGF).

Author Response

Dear reviewer, 

Thank you for your time to review my manuscript titled " Large Granular Lymphocytic Leukemia: Clinical Features, Molecular Pathogenesis, Diagnosis and Treatment". We appreciate the time and effort that you have dedicated to providing your valuable feedback on this manuscript. Below is a summary of the incorporated changes reflecting the suggestions provided by you. We have highlighted the changes within the manuscript.

Comment 1: line 339: Epidermal Growth Factor (EGF) should be as epidermal growth factor (EGF).

Response: Greatly appreciate for catching this. This was changed as per the suggestion. 

Sincerely, 

Dr. Ullah and the team